# HURP regulates Kif18A recruitment and activity to synergistically control microtubule dynamics

Juan M. Perez-Bertoldi [1,6], Yuanchang Zhao [2,3,6], Akanksha Thawani [3], Ahmet Yildiz [1,2,3,4] ✉ & Eva Nogales [3,4,5] ✉

During mitosis, microtubule dynamics are regulated to ensure proper alignment and segregation of chromosomes. The dynamics of kinetochore-attached microtubules are regulated by hepatoma-upregulated protein (HURP) and the mitotic kinesin-8 Kif18A, but the underlying mechanism remains elusive. Using single-molecule imaging in vitro, we demonstrate that Kif18A motility is regulated by HURP. While sparse decoration of HURP activates the motor, higher concentrations hinder processive motility. To shed light on this behavior, we determine the binding mode of HURP to microtubules using cryo-EM. The structure helps rationalize why HURP functions as a microtubule stabilizer. Additionally, HURP partially overlaps with the microtubule-binding site of the Kif18A motor domain, indicating that excess HURP inhibits Kif18A motility by steric exclusion. We also observe that HURP and Kif18A function together to suppress dynamics of the microtubule plus-end, providing a mechanistic basis for how they collectively serve in microtubule length control.

Proper segregation of genetic material during cell division relies on the organization of microtubule filaments into a bipolar spindle. Polarity and polymerization dynamics of spindle microtubules are regulated by a plethora of microtubule-associated proteins (MAPs) and molecular motors to form specialized sub-structures within the spindle. Kinetochore-fibers (K-fibers), made of parallel arrays of kinetochore-bound (K-microtubules), and non-kinetochore-bound microtubules, are an example of spindle specialization. During metaphase, the chromosomes attach to K-fibers and remain tightly bound to the kinetochores during the growth and shrinking phases of the dynamic plus-ends, while the spindle globally maintains a constant steady-state length[1–3]. Because improper chromosome alignment can lead to aneuploidy, cancer, and birth defects[4–7], it is important to understand the mechanisms regulating the properties and function of K-fibers. Yet, the molecular mechanism of how microtubule properties are altered

to robustly engage K-fibers with kinetochores throughout cell division is not well understood.

K-fibers recruit specific MAPs and motors to promote microtubule bundling and modulate their plus-end dynamics. Hepatoma-upregulated protein (HURP) is a spindle assembly factor that localizes to the chromatin-proximal region of K-fibers in a process mediated by Ran-GTP signaling[8–10]. Through its stabilizing and bundling activities, HURP regulates K-fiber dynamics and spindle morphology, contributing to chromosomal movements and alignment[11,12]. Both depletion and overexpression of HURP lead to defective spindles that cannot align chromosomes effectively[10,13]. However, how HURP binds to microtubules and stabilizes their ends remains unexplored.

K-fibers also recruit Kif18A, a member of the kinesin-8 family that walks towards and accumulates at the plus-ends of K-microtubules[14–17]. Kif18A modulates directional switching of chromosome oscillations

[1]Biophysics Graduate Group, University of California, Berkeley, CA, USA. [2]Physics Department, University of California, Berkeley, CA, USA. [3]Department of Molecular and Cell Biology, University of California, Berkeley, CA, USA. [4]Molecular Biophysics and Integrative Bioimaging Division, Lawrence Berkeley National Laboratory, Berkeley, CA, USA. [5]Howard Hughes Medical Institute, University of California, Berkeley, CA, USA. [6]These authors contributed equally: Juan M. Perez-Bertoldi, Yuanchang Zhao. ✉e-mail: yildiz@berkeley.edu; enogales@lbl.gov

and the relative motion of sister kinetochores, which are crucial for maintaining tension and for enabling the spindle assembly checkpoint to monitor and correct errors[18]. Functional studies showed that Kif18A performs these cellular roles by suppressing the plus-end growth of K-microtubules in a length-dependent manner[19–21], but the mechanism by which Kif18A regulates microtubule dynamics remains controversial[22]. An earlier study showed that Kif18A actively depolymerizes GMPCPP-stabilized microtubules in vitro[16], whereas another study argued that Kif18A primarily controls microtubule length by acting as a capping protein and restraining growth rate, without necessarily inducing depolymerization[23]. Studies in live cells suggested that the N-terminus of HURP binds to microtubules[24,25] and regulates Kif18A localization on these tubulin polymers[13]. Kif18A-depleted cells exhibit similar phenotypes to those of HURP-depleted and HURP-overexpressed cells[13,17,26], suggesting a relationship between the functions of these two proteins.

Although Kif18A and its homologs have been studied in vivo and in vitro, the molecular understanding of how it accumulates at the plus-end of microtubules and modulates microtubule dynamics together with HURP remains to be demonstrated. In this study, we use total internal reflection fluorescence (TIRF) microscopy to reveal how Kif18A motility on microtubules is influenced by HURP and how these two proteins work together to modulate plus-end dynamics of microtubules in vitro. We show that HURP promotes recruitment of Kif18A to microtubules and activation of Kif18A motility. Yet, at high HURP concentrations, we observe antagonism between HURP and Kif18A. To better understand this mutual antagonism, we employed cryogenic electron microscopy (cryo-EM) to visualize how HURP and Kif18A bind to the microtubules. Our structural studies reveal an overlap of the binding surfaces of the two proteins on the microtubule, resulting in the inhibition of KIF18A by excess HURP on the microtubule. We also show that the interplay between HURP and Kif18A at the plus-end modulates microtubule dynamics, providing a mechanistic explanation for how these proteins contribute to K-microtubule stabilization and length control.

## Results

### HURP regulates Kif18A motility in a concentration-dependent manner

To explore how HURP regulates Kif18A motility, we recombinantly expressed full-length and truncated human HURP constructs C-terminally tagged with an enhanced green fluorescent protein (HURP[1-173]-eGFP, HURP[1-285]-eGFP, HURP[1-400]-eGFP and HURP[1-846]-eGFP (full-length HURP)) (Fig. 1a, Supplementary Fig. 1). We used TIRF microscopy to quantify microtubule binding of HURP (Fig. 1b) and showed that all four HURP constructs bound to microtubules (Fig. 1c). The half-maximal saturation concentration ($K_d$) of HURP[1-285] binding to taxol-stabilized microtubules was ~0.3 μM at physiological salt concentration (150 mM) (Fig. 1d).

We next assayed the motility of human Kif18A on surface-immobilized microtubules in the presence or absence of near-saturating HURP concentration (1 μM) (Fig. 1b). HURP[1-173] and HURP[1-285] substantially enhanced the run frequency of Kif18A without significantly altering its velocity or run time (Fig. 1e, f), suggesting that HURP contains a Kif18A-activating site between amino acids 1-173, referred to as the "activating motif". The stimulatory effect of HURP on Kif18A motility appeared specific, as HURP[1-285] did not activate kinesin-1 Kif5B (Supplementary Fig. 2), and Kif18A activation was not observed with the MAPs doublecortin (DCX), MAP7, or tau (Supplementary Fig. 3). HURP[1-400] and HURP[1-846] also increased the run frequency of Kif18A motors, but the effect was not as pronounced as with the shorter HURP constructs. This lesser effect is because HURP[1-400] and HURP[1-846] reduced Kif18A velocity several-fold, resulting in more motors being counted as stationary (Fig. 1e, f).

We then investigated how titration of full-length HURP[1-846] affects Kif18A motility. Kif18A run frequency exhibited a biphasic behavior, with an initial activation phase followed by a decrease in the run frequency at near saturating concentrations (1 μM) of HURP[1-846]. In comparison, Kif18A velocity displayed a steady decrease, and the run time steadily increased under increasing concentrations of HURP[1-846] (Fig. 2).

We next removed the C-terminal tail of Kif18A (Kif18A[1-480], Fig. 1a, Supplementary Fig. 1) to explore the activation and HURP-mediated regulation of Kif18A motility in its absence. Interestingly, in the absence of HURP, the tail-truncated Kif18A[1-480] exhibited significantly more frequent runs than full-length Kif18A, suggesting that this motor is partially autoinhibited by its tail, analogous to other kinesins[27–31]. HURP[1-846] reduced the velocity of Kif18A[1-480], suggesting that the slow-down effect does not involve the C-terminal tail of Kif18A. Notably, Kif18A[1-480] run frequency did not exhibit the activation phase observed with the full-length motor. Instead, its run frequency, velocity, run time, and run length decreased as HURP[1-846] concentration was increased (Fig. 2, Supplementary Fig. 4).

We also analyzed the motility of full-length Kif18A and Kif18A[1-480] under 0–6.5 μM HURP[1-285]. The addition of this shorter HURP construct resulted in up to a seven-fold increase in Kif18A run frequency in a concentration-dependent manner. On the other hand, HURP[1-285] reduced the run time of Kif18A at higher concentrations, which is consistent with HURP[1-278] overexpression mimicking a phenotype of Kif18A depletion in vivo[13]. HURP[1-285] binding to microtubules did not significantly change the run frequency or velocity of Kif18A[1-480], but substantially decreased its run time (Fig. 2). Collectively, our functional studies suggest that HURP interacts with full-length Kif18A, recruiting the motor to the microtubules, and rescuing it from autoinhibition to activate its motility. The 285-400 segment of HURP acts as a "decelerating motif" that reduces the speed of Kif18A motility by 90% at 1 μM HURP. Since the effect is observed for the motor even in the absence of its tail, this HURP segment may interact with a region constrained to amino acids 1-480 of Kif18A. This idea is further supported by in silico AlphaFold 3[32] models of Kif18A[1-480] in the presence of HURP[285-400], showing a robustly predicted interface between HURP[321-339] and a region around the central β-blades of Kif18A's motor domain (Fig. 3a, Supplementary Fig. 5). Unlike the 1-173 segment of HURP that appears to release auto-inhibition and activate motility, this interaction markedly reduces Kif18A's velocity.

We next performed dual-color imaging of KIF18A and HURP in motility assays to directly observe whether HURP can bind to KIF18A and be carried to the microtubule plus end. In this assay, a high concentration of Kif18A (300 nM) and a low concentration of HURP (25 nM) were used to drive the formation of a Kif18A-HURP complex while resolving single HURP molecules on microtubules at the diffraction-limited resolution of fluorescence imaging. Full-length Kif18A was able to carry all of the tested HURP constructs, which contain the "activating motif". In comparison, Kif18A[1-480] only carried HURP[1-400] and HURP[1-846], which contain the "decelerating motif" but lack the "activating motif" interaction in the absence of the kinesin tail. Only the HURP[1-285]-Kif18A[1-480] pair did not exhibit directional transport of HURP, since the 1-173 "activating motif" cannot act in the presence of a tail-less motor, and the 285-400 "decelerating motif" is missing in this HURP construct (Fig. 3b, c).

### HURP bridges tubulin subunits across protofilaments

To further understand the inhibition of Kif18A motility observed at higher HURP concentrations, we visualized how HURP interacts with the microtubule surface. A previous study identified two distinct microtubule-binding domains (MTBDs) on the N-terminus of HURP: MTBD1 is the constitutive, high-affinity interaction site (HURP[105-150]), whereas MTBD2 (HURP[22–50]) has weaker microtubule affinity and is

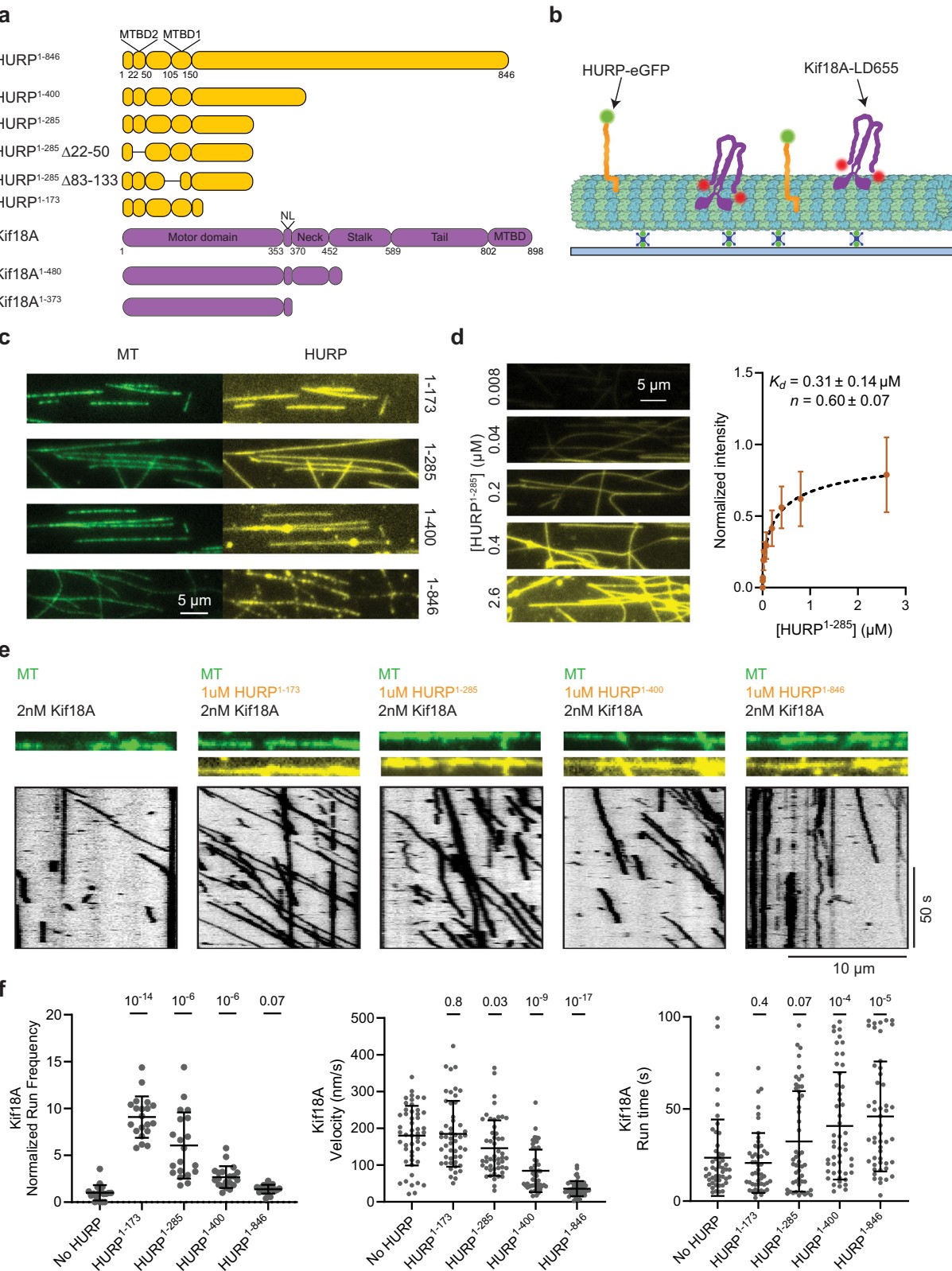

regulated by importin-β[24] (Fig. 1a). Our TIRF imaging assays revealed that HURP[1-285] binds to taxol-stabilized microtubules with an order of magnitude lower $K_d$ (-0.029 μM) in the absence of 150 mM added salt (see Methods, Figs. 1d and 4a, b), indicating that HURP's affinity for microtubules is salt-dependent, likely due to the shielding of electrostatic interactions[33]. We next used cryo-EM to determine the structure of HURP[1-285] bound to taxol-stabilized microtubules under saturating conditions. After following a seam-determination

procedure[34] and exploiting the pseudo-symmetry of the microtubule, we generated a 3.1 Å cryo-EM density map (Fig. 4c, Supplementary Figs. 6 and 7a) that allowed us to manually model HURP MTBD1 de novo, identifying residues 87-132 within the constitutive binding site. No density corresponding to the HURP MTBD2 was present in the map (Supplementary Fig. 8), indicating that the interaction between this region and microtubules likely involves flexible elements in HURP or tubulin.

**Fig. 1 | HURP contains different elements that can activate or decelerate Kif18A motility. a** Domain organization of full-length HURP and Kif18A, and different HURP and Kif18A truncations used in this study (NL: neck-linker). **b** Schematic of the in vitro reconstitution of Kif18A motility on surface-immobilized microtubules (MT) in the presence of HURP[44]. **c** Representative fluorescence images showing microtubule binding of HURP constructs used for motility assays. All HURP constructs were tested at 1 μM concentration. **d** (Left) Representative images showing HURP[1-285] binding to microtubules at different concentrations. (Right) Quantification of HURP[1-285] binding to microtubules. The center circle and whiskers represent the mean and standard deviation (S.D.), respectively. $K_d$ was determined from a fit to a Hill equation (dashed

curve, $N = 20$ microtubules for each condition). **e** Representative kymographs showing motility of 2 nM Kif18A in the presence of 1 μM of different HURP constructs. Microtubule and HURP signals are shown in green and yellow, respectively. Representative kymographs are from the same microscopy session, with the same batches of proteins. **f** Normalized run frequency ($N = 20$ kymographs for each condition), velocity, and run time ($N = 50$ motors for each condition) of 2 nM Kif18A in the presence of different HURP constructs. The center line and whiskers represent the mean and S.D., respectively. $P$ values were calculated from a two-tailed $t$-test, compared to the no HURP condition. The in vitro motility assays were performed with 3 technical replicates. Source data are provided as a Source Data file.

We observed two structural motifs for MTBD1. An α-helical density (HURP[87-114]) spans laterally across β-tubulin and establishes contacts (via L94, Y97, K98, and K101) with residues on the β-tubulin H12 helix (E410 and M406) involving both hydrogen bonding and hydrophobic interactions (Supplementary Fig. 9a–c). Additionally, an extended loop, including residues 115-132, inserts in the inter-protofilament groove and contacts two laterally adjacent tubulins, thus stapling the protofilaments together. These interactions are mainly driven by hydrophobic residues that insert in hydrophobic pockets on the tubulin subunits (Fig. 4d, e, Supplementary Fig. 9d, e). Most of the HURP residues that participate in these interactions are highly conserved among species (Supplementary Fig. 10). The deep insertion in the inter-protofilament groove and the bridging interactions between adjacent tubulin subunits are consistent with HURP's role as a microtubule-stabilizing factor[10,12]. Additionally, we observed a slight (0.9 Å) compaction of spacing between adjacent tubulin dimers along a protofilament with respect to a taxol-microtubule lattice in the absence of HURP, raising the possibility that this protein could indirectly alter the recruitment of KIF18A and other factors by modifying the structural properties of the lattice. The microtubule bound structure of the MTBD1 of HURP resembles that of another spindle assembly factor, TPX2, which also contains a dual binding motif that establishes lateral and longitudinal contacts to staple tubulins together[35] (Fig. 4f). This similarity could point to a shared molecular mechanism for these two critical players in microtubule stabilization for the regulation of K-fiber dynamics (see Discussion).

To better understand how MTBD1 and MTBD2 contribute to microtubule binding, we designed truncated versions of HURP[1-285], either lacking the structurally ordered segment visualized in our density map (HURP[1-285] Δ83-133) or the unstructured MTBD2 (HURP[1-285] Δ22-50), and tested their microtubule binding. As expected, removal of either MTBDs decreased the affinity of HURP for microtubules, with the largest impact observed upon removal of the structured MTBD1 (Supplementary Fig. 11). We also uncovered a complex HURP-microtubule binding behavior, likely involving both structured and unstructured elements in HURP and tubulin. These interactions may influence binding cooperativity, as indicated by differences we observed in cooperative binding of HURP constructs to microtubules, and by how HURP compacts the lattice geometry (Fig. 1a, Supplementary Figs. 1 and 11).

## HURP and Kif18A cannot simultaneously occupy the same tubulin dimer

Superimposing our HURP-microtubule model with a previously reported Kif18A-microtubule structure (PDB 5OCU)[36] indicated a potential steric clash between the α-helical segment of HURP and the L8/β5 tubulin binding motif of the Kif18A motor domain. Such overlap would be consistent with our observation that Kif18A cannot walk processively towards the plus-end at high HURP concentrations. To test whether HURP and Kif18A can co-occupy the microtubule lattice, we generated a monomeric Kif18A construct containing the motor domain and neck linker (Fig. 1a, Supplementary Fig. 1) fused to SNAP-tag (Kif18A[1-373]-SNAP). Then, we used TIRF imaging to determine the co-decoration of microtubules with HURP[1-285]-eGFP and Kif18A[1-373]-SNAP

labeled with an LD655 dye (Fig. 5a). The two proteins coated the microtubules with similar surface densities when mixed at equal concentrations. Microtubule-binding of HURP[1-285] and Kif18A[1-373] was negatively correlated (Pearson's $r = −0.83$, Fig. 5b), suggesting that HURP and Kif18A compete against each other for the available binding sites on the microtubule.

To directly determine whether HURP and Kif18A exclude each other on microtubules, we determined the structure of microtubules co-decorated with HURP[1-285] and Kif18A[1-373]. To increase the likelihood of these proteins occupying the same or adjacent tubulins, we decorated taxol-stabilized microtubules with equimolar and near-saturating concentrations of HURP and Kif18A. Processing of the cryo-EM images generated a consensus reconstruction for the co-decorated microtubule, showing clear density features for both HURP and Kif18A (Fig. 5c). The symmetry-expanded particle set was further refined using a mask englobing a single tubulin dimer and the density corresponding to single copies of HURP and Kif18A. This density map, which corresponds to an average, clearly shows a steric clash between the expected structural elements from HURP and Kif18A (Fig. 5d). To dissect the different populations that could be contributing to the reconstruction, we proceeded with alignment-free 3D classification, which detected three distinct classes (Fig. 5e). The first class contained density for tubulin and HURP (~40% of the particles, 3.0 Å), the second class featured tubulin and Kif18A (~39% of the particles, 3.1 Å), while the third class showed tubulin density only (~21% of the particles, 3.5 Å) (Supplementary Figs. 7b and 12). For the Kif18A-containing class, we were able to build a model (Supplementary Fig. 13a, b) and visualize kinesin-tubulin interactions (Supplementary Fig. 13c, d).

Further classification of the Kif18A-containing particles with a mask around the inter-protofilament groove did not show a reliable class where Kif18A displaced HURP's α-helix but the groove-binding loop was still engaged. This result further confirms an antagonistic binding mode, where the presence of the motor domain of Kif18A is incompatible with HURP's MTBD1 engaging the microtubule, since the motor displaced both the α-helix and the groove-binding loop of HURP from tubulin. We concluded that HURP and the motor domain of Kif18A cannot occupy the same tubulin dimer on the microtubule lattice due to steric exclusion.

## Human Kif18A adopts a pro-motility state on microtubules

Our microtubule-bound structure of Kif18A provides insight into the nucleotide-dependent conformation and depolymerizing activity of Kif18A. In *C. Albicans* kinesin-8 Kip3, loop 2 controls whether the motor adopts a pro-motility or pro-depolymerization state by establishing specific contacts with α-tubulin on curled ends of protofilaments[37]. In the structure of Kip3 bound to a non-hydrolyzable ATP analog (AMP-PNP) that mimics the ATP-bound state (PDB 7TQX), loop 2 is clearly resolved and stabilized by α-tubulin, the neck linker is undocked, and the nucleotide-binding pocket is open. In the post-hydrolysis ADP.Pi state, mimicked by ADP-AlFx, (PDB 7TQY), the nucleotide pocket closes, the neck linker docks, and Kip3 adopts a pro-motility state. In comparison, the loop 2 region of Kif18A was disordered in the cryo-EM density (see Supplementary Fig. 14a for a comparison of our Kif18A model with that of Kip3 (PDB 7TQX) aligned on β-tubulin). Our

structure of the Kif18A motor bound to AMP-PNP has a docked neck linker and the nucleotide-binding pocket adopts a semi-closed conformation, which more closely resembles that of Kip3 ADP-AlFx than that of Kip3 AMP-PNP (Supplementary Fig. 14b, c). Recent work has shown that this transition in Kip3 is driven by changes in the interaction between loop 2 and α-tubulin[37], suggesting that Kif18A adopts a

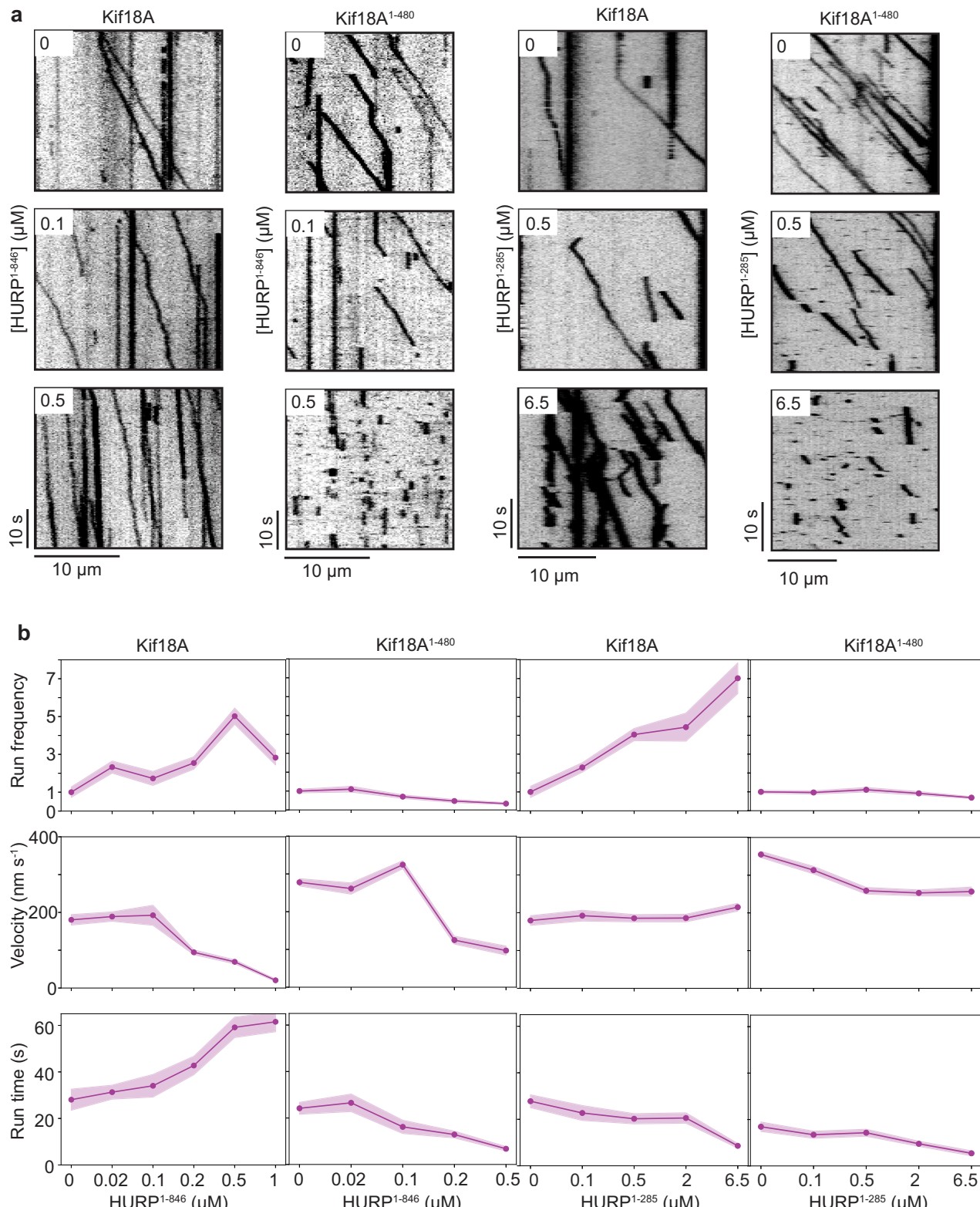

**Fig. 2 | HURP differentially affects Kif18A motility in a concentration-dependent manner. a** Representative kymographs showing the motility of full-length Kif18A and Kif18A[1-480] in the presence of 0, 0.1, and 0.5 μM HURP[1-846] and 0, 0.5, and 6.5 μM HURP[1-285]. **b** Normalized run frequency, velocity, and run time of Kif18A and Kif18A[1-480] in the presence of different concentrations of HURP[1-285] or HURP[1-846]. For run frequency, $n$ = 10 microtubules for each condition. For velocity and run time, $N$ = 32, 50, 33, 48, 50, 40, 25, 25, 25, 25, 25, 52, 44, 54, 52, 52, 52, 51, 52, 52, 52 motors, from left to right. In (**b**), the line and shadows represent the mean and standard error (S.E.), respectively. The in vitro motility assays were performed with 2 technical replicates. Source data are provided as a Source Data file.

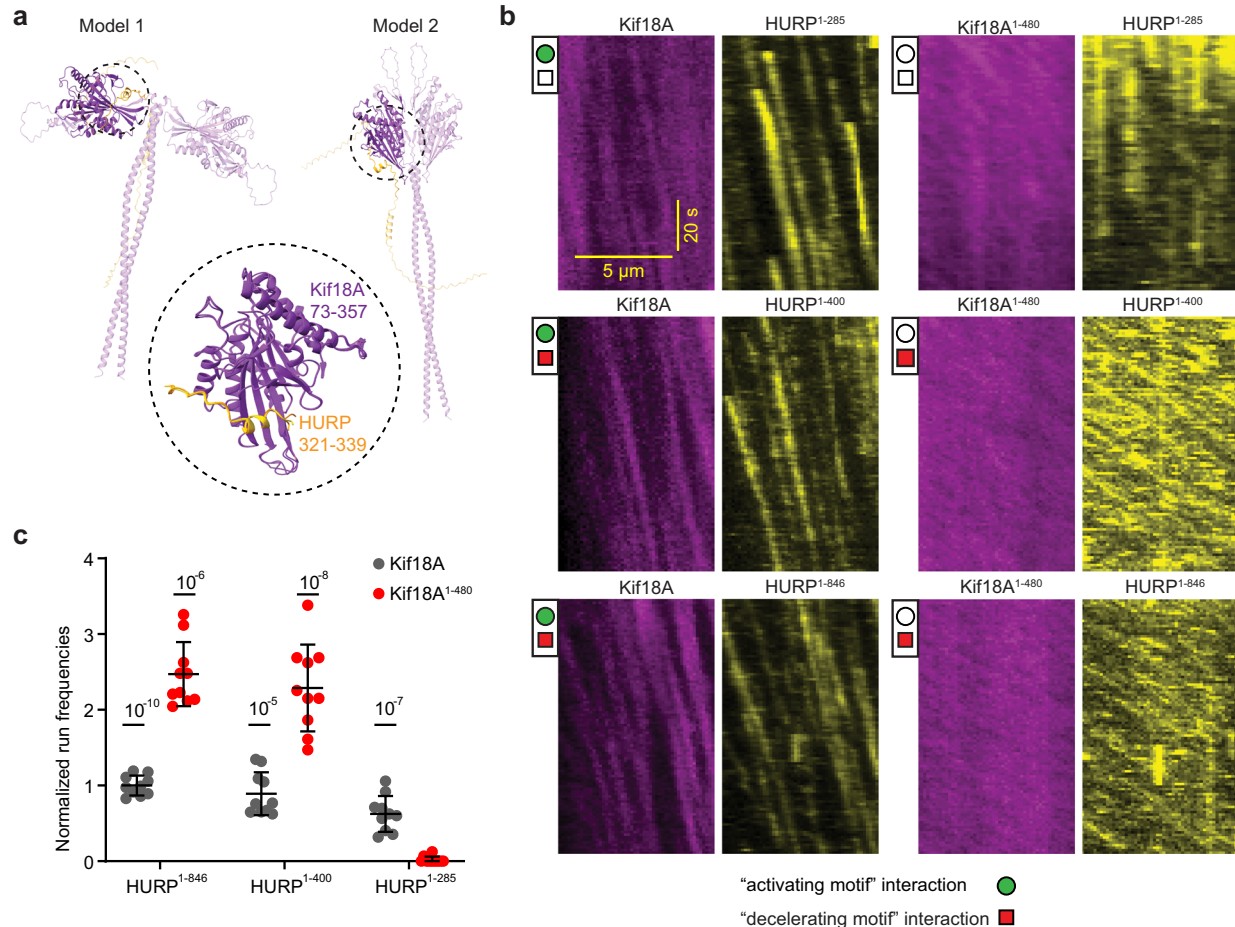

**Fig. 3 | Kif18A and HURP interact with each other at two distinct sites.**
**a** Examples of AlphaFold 3[32] in silico predicted models of Kif18A[1-480]-HURP[285-400] showing flexibility of the two motor heads and a preserved Kif18A-HURP interaction highlighted inside dashed circles. The conserved Kif18A-HURP interface was present in 3 out of 5 models produced by the software. Encircled region highlights the alignment of 3 predicted models on the motor head that exhibits the HURP interaction. Kif18A[73-357] is shown in purple in all 3 models, while HURP[321-339] is displayed in varying shades of orange in the different models. **b** Representative kymographs of

300 nM full-length Kif18A or truncated Kif18A[1-480] motors in the presence of 25 nM HURP[1-285], HURP[1-400] or HURP[1-846]. **c** Normalized run frequency of different HURP constructs that were carried to the plus-end by Kif18A or Kif18A[1-480]. Frequencies were normalized to HURP[1-846] carried by Kif18A. The center line and whiskers represent the mean and S.D., respectively. *P* values were calculated from a two-tailed *t*-test, compared to HURP[1-285] carried by Kif18A[1-480] (*N* = 10 microtubules for each condition). Source data are provided as a Source Data file.

pro-motility conformation in the ATP-bound state. A sequence alignment of loop 2 shows that key residues interacting with α-tubulin in *C. Albicans* Kip3 have diverged in the human homolog (Supplementary Fig. 14d)[38], highlighting how sequence divergence in this region could be driving differences in the mechanism and function of kinesin-8 motors across species.

## HURP and Kif18A synergistically control microtubule length

We next turned our attention to determine how HURP and Kif18A affect microtubule dynamics. Consistent with earlier reports[14–16], in the absence of free tubulin, 0.1 μM Kif18A led to ~1 nm/s depolymerization of GMPCPP-stabilized microtubules. This depolymerization rate is 4-fold faster than the spontaneous depolymerization of GMPCPP-microtubules in the absence of Kif18A, but an order of magnitude slower than depolymerization of microtubules by kinesin-13s[39]. The addition of 1 μM HURP[1-173] mitigated the activity of Kif18A, reducing depolymerization rates to ~0.05 nm/s (Supplementary Fig. 15). We next investigated microtubule dynamics under conditions that included Kif18A with or without HURP. When Kif18A was added to the polymerization mixture alongside free tubulin, the run frequency of motors was significantly reduced (Supplementary Fig. 16), likely due to the interaction of Kif18A with free tubulin. To circumvent this issue and

ensure attachment of Kif18A to the microtubules, we pre-incubated Kif18A with microtubule seeds in the presence of AMP-PNP. We next flowed free tubulin, GTP, and ATP into the chamber to induce microtubule polymerization and Kif18A motility and monitored microtubule growth. Upon introduction of ATP, Kif18A started to walk and accumulated at the plus-end of the microtubule (Fig. 6a, b). We noticed a significant decrease in the plus-end growth velocity of microtubules with the accumulation of Kif18A, whereas the minus-end growth remained unaffected. The Kif18A puncta at the plus-end typically released in a single step, and the growth at the plus-end resumed immediately upon Kif18A's release (Fig. 6a, b). These results are consistent with the idea that Kif18A acts as a 'molecular cap' that hinders plus-end growth[23,40]. This capping effect was found to rely on the C-terminal region of Kif18A, since Kif18A[1-480] failed to exhibit microtubule capping, despite its robust motility on dynamic microtubules (Fig. 6c–e).

In the absence of Kif18A, addition of HURP[1-400] to dynamic microtubules increased the rescue frequency, without affecting microtubule growth, shrinking rates[12] or the catastrophe frequency (Fig. 6f, g). These results led us to investigate HURP's possible effect on the capping mechanism exerted by Kif18A. The introduction of HURP[1-400] prolonged Kif18A's motility on microtubules (Fig. 1e, f) and

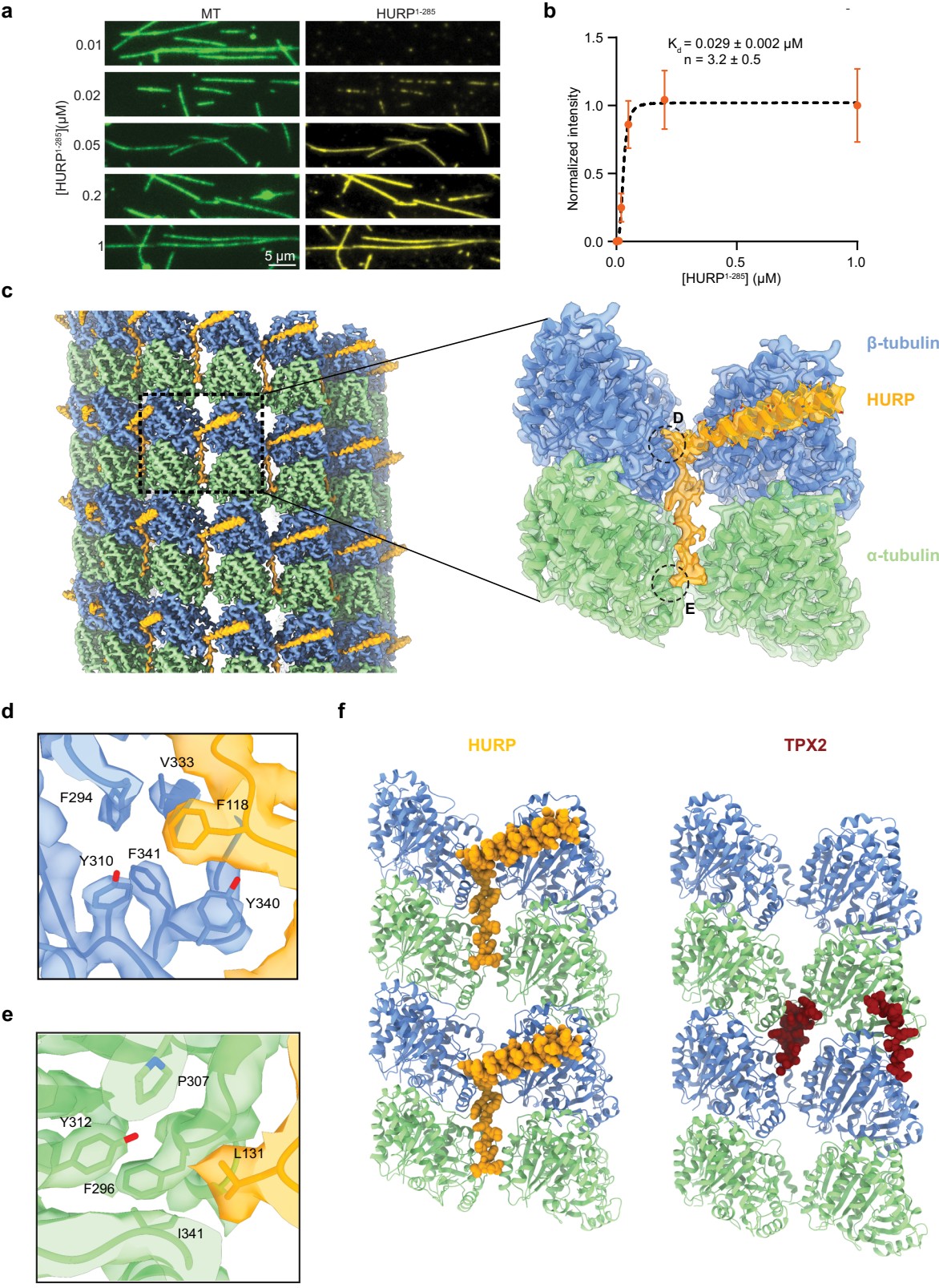

its retention at the plus-end, thereby extending the duration at which the Kif18A cap inhibits the growth of the plus-end. We also found that HURP$^{1-400}$ co-migrated with Kif18A and accumulated at the plus-end (Fig. 7a, b, Supplementary Fig. 17a). HURP$^{1-285}$ also extended Kif18A's capping period (Fig. 7b, c, Supplementary Fig. 17b), but the effect was less pronounced, possibly because this construct lacks the "decelerating motif" and does not significantly increase the residence time of

Kif18A on microtubules (Figs. 1 and 2). In this case, we observed multiple events where the microtubule initially exhibits growth from the seed, followed by the capping of the plus-end by Kif18A and a reduction of growth velocity. Subsequently, normal growth resumes upon detachment of the motor from the microtubule plus-end (Fig. 7d, e, Supplementary Fig. 17c). Collectively, our findings indicate that HURP and Kif18A synergistically regulate microtubule length. Plus-end

**Fig. 4 | HURP interacts with microtubules through a dual binding site.**
**a** Representative images showing microtubule decoration with different concentrations of HURP[1-285] in the absence of added salt. **b** Quantification of HURP[1-285] binding to microtubules. The center line and whiskers represent the mean and S.D., respectively. The $K_d$ is determined from a fit to a Hill equation (dashed curve, $N = 50$ microtubules for each condition). Source data are provided as a Source Data file. **c** (Left) Surface representation of the symmetrized microtubule-HURP cryo-EM density map generated with a mask around the entire microtubule. α-tubulin, β-tubulin, and HURP are shown in green, blue and orange, respectively. The boxed region marks an area including two tubulin dimers and one HURP molecule. (Right) Final microtubule-HURP symmetry-expanded map generated with a mask focusing on two neighboring tubulin dimers. A single HURP molecule is shown for clarity. The refined model is shown in ribbon representation and the map displayed with transparency. HURP side chains are displayed with atom representation (orange: C, red: O, blue: N, yellow: S). **d, e** Details of the interactions between HURP and tubulin at the inter-protofilament groove. **f** Comparison between HURP (orange, this study) and TPX2 (dark red, PDB 6BJC) microtubule-bound structures.

accumulation of Kif18A arrests microtubule growth. Kif18A also slowly depolymerizes microtubules, but this effect is reduced by the stabilizing activity of HURP. Additionally, Kif18A caps the microtubule plus-end against shrinkage, and this capping activity is enhanced by HURP. Our results show that the combination of HURP and Kif18A substantially suppresses the plus-end dynamics of microtubules and thus maintains a constant microtubule length.

## Discussion

In this work, we show that HURP can recruit and activate kinesin-8 Kif18A on microtubules, providing insight into how these two proteins organize K-fibers and control their length. Our observations indicate that HURP concentration can be fine-tuned to produce different outcomes on Kif18A motility. At physiologically relevant regimes (i.e., 0.32 μM HURP was reported in *Xenopus laevis* egg extracts[41]), HURP activates Kif18A, presumably by releasing an auto-inhibitory interaction between the motor domain and the C-terminal tail of kinesin. This autoinhibitory mechanism is conserved in other kinesin families, including kinesin-1, kinesin-3, and kinesin-7, and provides a way to locally regulate the engagement of these motors with microtubules[27–31].

Our findings suggest that HURP could interact with Kif18A through two distinct sites: The 1-173 segment of HURP contains the "activating motif" that affects the interaction between the motor and tail domains of Kif18A, releases auto-inhibition, and activates Kif18A motility. In comparison, the 285-400 segment of HURP contains the "decelerating motif" that interacts with the 1-480 segment of Kif18A and markedly reduces Kif18A's velocity (Fig. 3). Interestingly, a crosstalk between the N and C-termini of HURP has been described in previous studies, showing that HURP phosphorylation in the C-terminus by the Aurora A kinase can regulate accessibility of its N-terminus and contribute to HURP localization[25,42,43]. Therefore, the C-terminus of HURP may also regulate the activating role of the HURP N-terminus, resulting in fewer and slower Kif18A runs on the microtubule. Future studies will be required to address the roles of these two regions of HURP, and their post-translational modifications, on Kif18A motility.

During metaphase, HURP forms a comet-like gradient along K-microtubules, driven by the local concentration of Ran-GTP[11,43]. HURP-mediated activation could promote the enrichment of Kif18A in chromatin-proximal regions of the K-fibers. However, increased recruitment of motors to microtubules does not lead to productive motility at higher HURP concentrations, suggesting that concentration-dependent effect of HURP on kinesin motility may help fine-tune the accumulation of Kif18A to the plus ends of K-fibers. We previously reported a similar regulatory role of MAP7 in kinesin-1 motility[44], underscoring that concentration-dependent regulation of motors by MAPs could serve as a general mechanism for spatial and temporal regulation of microtubule-driven processes.

The biphasic regulation of Kif18A by HURP concentration we observed in vitro can explain why the phenotypes for HURP depletion and overexpression resemble each other in vivo. When HURP is depleted, Kif18A likely remains in an auto-inhibited state, limiting the number of landing events that result in processive motility. On the other hand, when HURP is overexpressed in pathological states of the cell[8,45–47], it could saturate the microtubule surface and block efficient Kif18A walking. Both of these situations would lead to inefficient accumulation of Kif18A at the kinetochore-proximal end, promoting defects in chromosome congression during mitosis (Fig. 8a).

In kinesin-8 Kip3p, the yeast homolog of Kif18A, the length-dependent accumulation on the plus-end of K-fibers has been attributed to kinesins randomly landing on the microtubule surface and processively walking towards the kinetochore-proximal end, where they would accumulate and quench polymerization dynamics. This "antenna model" explained how members of the kinesin-8 family could depolymerize longer microtubules at a faster rate than shorter ones in organisms like yeast that lack HURP orthologs[19,22,48]. We suggest that productive landing of Kif18A might not occur randomly in vivo in higher eukaryotes containing HURP or HURP-related proteins, but instead be more prevalent in regions where HURP localizes and activates the motor, closer to the plus-end of K-microtubules, thus facilitating its accumulation. Such mechanism could play a synergistic role with that proposed in the "antenna model" to fine-tune Kif18A localization in K-fibers.

Cryo-EM imaging of microtubule-bound HURP revealed how this MAP associates with microtubules through a bipartite binding motif that stabilizes adjacent tubulin dimers by forming lateral contacts. Consistent with its role as a stabilizer, HURP has been implicated in attenuating microtubule dynamic instability by increasing the rescue frequency in vitro[12,49]. Our structural model suggests that this function could be exerted by laterally stabilizing protofilaments, and attenuating their peeling at the microtubule tip during catastrophes, therefore increasing the likelihood of rescues[50,51].

The microtubule binding mechanism of HURP resembles that of another spindle assembly factor and stabilizer, TPX2[35,52,53]. Both HURP and TPX2 are regulated by the Ran-GTP pathway[10,54], both interact with mitotic kinesins (HURP with Kif18A and Kif11/Eg5[13,43] and TPX2 with Kif11/Eg5 and Kif15[55–57]), and both nucleate and stabilize K-microtubules[10,42,58–61]. The structured binding motif in HURP stabilizes lateral interactions, while TPX2 establishes both lateral and longitudinal contacts. A recent structural and functional study focusing on microtubule nucleation proposes that HURP and TPX2 work in tandem to enhance this process on K-fibers[62]. The structural model in that study is consistent with ours, identifying the same microtubule binding motif for HURP. Intriguingly, TPX2 has been shown to interact with HURP and other partners for bipolar spindle formation[9], which could point to distinct but potentially complementary roles in assembling K-fibers.

Finally, we demonstrate that Kif18A localizes to the plus-ends of microtubules and caps their growth, likely through a mechanism that relies on the C-terminal tail of the motor. Given the previous identification of a non-motor MTBD at the C-terminal of Kif18A[40,63], this region may significantly influence the capping process. It remains to be demonstrated whether the C-terminal MTBD directly tethers to the plus-end and blocks the addition of new tubulin units, or, if it facilitates Kif18A's accumulation and retention at the plus-end by increasing its dwell time.

HURP enhances Kif18A's capping effect by increasing the residence time of the motor at the microtubule plus-end, in addition to

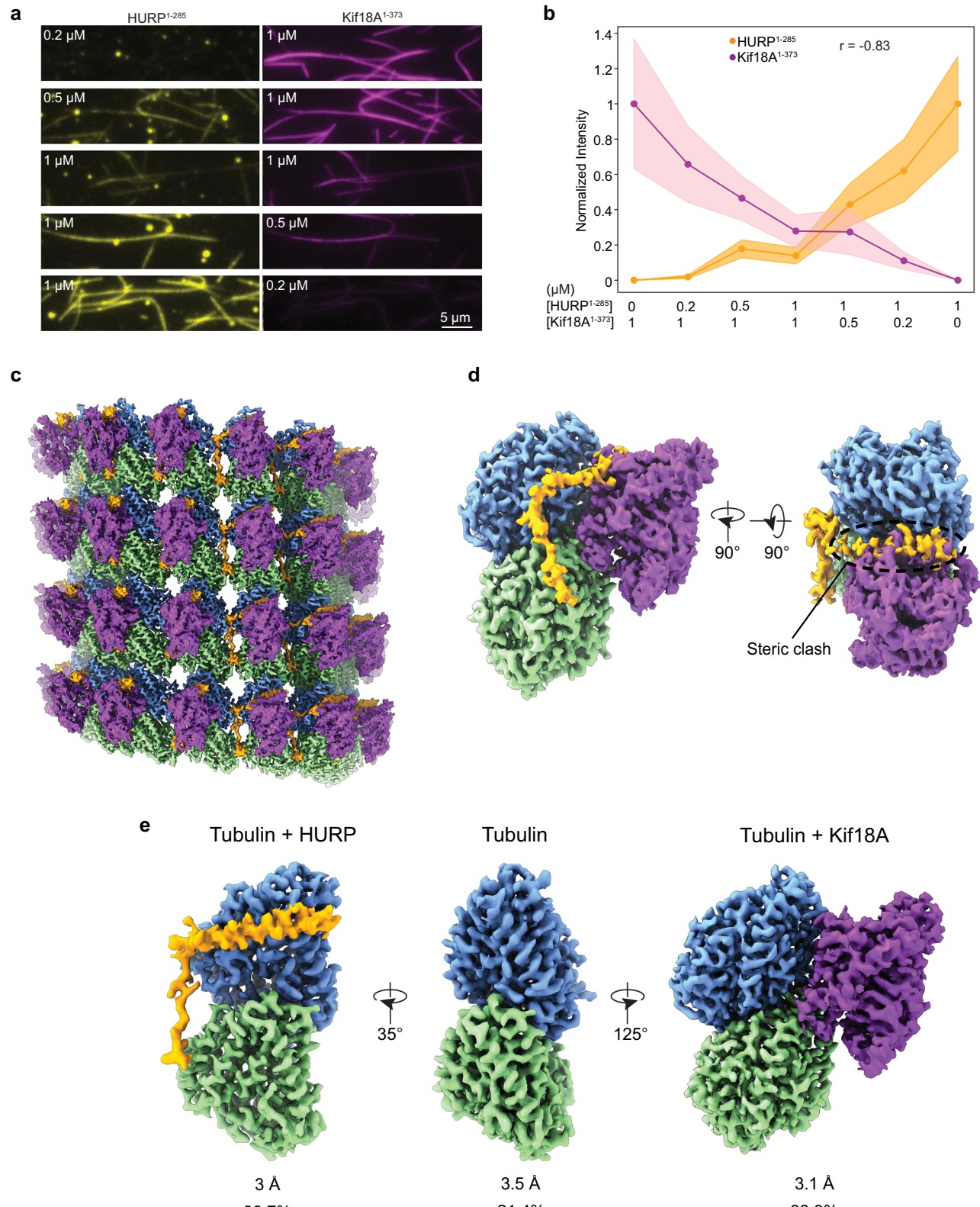

having its own role as a stabilizer. We propose that HURP and Kif18A work synergistically to reduce microtubule dynamics, which could serve as a spindle length control mechanism during mitosis (Fig. 8b). Our observation that HURP can be transported towards the plus-end by Kif18A has been reported for other MAPs[64] and suggests that this could play a secondary role in creating a HURP gradient, besides the canonical Ran-mediated pathway that dictates HURP distribution. Recent studies have reported that HURP shows differential binding to microtubules of variable length, through a mechanism that is still unclear and could involve centrosomal regulation[65]. Future studies will be required to reveal how HURP "senses" K-fiber length and tunes its dynamics together with Kif18A.

**Fig. 5 | HURP binding site partially overlaps with Kif18A's motor domain on the microtubules. a** Representative images of HURP$^{1-285}$-eGFP and Kif18A$^{1-373}$-SNAP binding to microtubules for different ratios of the two proteins. **b** Normalized fluorescence intensity for HURP$^{1-285}$ and Kif18A$^{1-373}$ for the protein ratios used in (**a**). The center line and shadows represent the mean and S.E., respectively (N = 50 microtubules for each condition; *r*: Pearson's correlation coefficient). Source data are provided as a Source Data file. **c** Surface representation of the symmetrized microtubule-HURP-Kif18A$^{1-373}$ cryo-EM density map generated with a mask around the entire microtubule. α-tubulin, β-tubulin, HURP, and Kif18A$^{1-373}$ are shown in green, blue, orange, and purple, respectively. **d** (Left) Final consensus reconstruction of the symmetry-expanded dataset generated with a mask focusing on two neighboring tubulin dimers (only one shown for clarity) and the bound HURP and Kif18A densities. (Right) Rotated volume showing the steric clash between HURP and Kif18A. The volumes are color coded as in (**c**). **e** Refined classes produced during alignment-free 3D classification are shown in different orientations for clarity. Resolution and particle class distribution for each are indicated.

## Methods

### Protein expression, purification, and labeling

HURP constructs were cloned from U2OS-derived cDNA and either inserted into a pRSFDuet-SUMO vector for bacterial expression, or into a TwinStrep-pFastBac vector for insect cell expression through a sequence and ligase-independent cloning strategy. Kif18A constructs were produced from the pMX229 Addgene plasmid deposited by Linda Wordeman. Linear inserts containing Kif18A residues 1-373, 1-480, and 1-898 were PCR-cloned from pMX229 and inserted into a pRSFDuet-SUMO vector carrying the coding sequence for a C-terminal SNAP tag as described above. The sequence of all constructs was verified either by Sanger or full-length plasmid sequencing.

Kif18A (full-length Kif18A-SNAP, Kif18A$^{1-480}$-SNAP, Kif18A$^{1-373}$-SNAP) and truncated HURP constructs (HURP$^{1-173}$-eGFP, HURP$^{1-285}$, HURP$^{1-285}$-eGFP, HURP$^{1-285}$ Δ22-50-eGFP, HURP$^{1-285}$ Δ83-133-eGFP, HURP$^{1-400}$-eGFP) were transformed to Rosetta2(DE3) competent cells, plated for kanamycin selection, and a single colony was grown in LB+kanamycin at 37 °C until OD reached 0.6. Expression was induced with 0.2 mM IPTG at 37 °C and cells were harvested after 4 h. Full-length HURP$^{1-846}$-eGFP was produced in Sf9 cells through baculovirus infection as previously described[12].

Cell pellets of HURP$^{1-173}$-eGFP, HURP$^{1-285}$, HURP$^{1-285}$-eGFP, and HURP$^{1-400}$-eGFP were lysed through sonication in His lysis buffer (20 mM HEPES pH 7.5, 300 mM KCl, 1 mM MgCl$_2$, 20 mM imidazole, 0.1% Tween-20, 10 mM BME, 1x benzonase, 1 protease inhibitor tablet) and the lysate was cleared by centrifugation at 18,000 rcf for 1 h at 4 °C. The cleared lysate was incubated with 2 mL of Ni-NTA agarose resin, previously equilibrated in lysis buffer, for 2 h at 4 °C. The resin was washed with 150 mL buffer, alternating washing buffer (20 mM HEPES pH 7.5, 300 mM KCl, 1 mM MgCl$_2$, 20 mM imidazole, 0.1% Tween-20, 10 mM BME) with washing buffer supplemented with 700 mM NaCl. The protein was then eluted overnight in washing buffer supplemented with the Ulp1 protease to cleave the N-terminal His-SUMO tag on the constructs. The eluted protein was diluted 4× with IEX A buffer (20 mM HEPES pH 7.5, 100 mM NaCl, 10 mM BME) and loaded onto a 5 mL HiTrap SP HP column for ion exchange chromatography. The protein was then eluted with a salt gradient from 0 to 50% IEX B buffer (20 mM HEPES pH 7.5, 2 M NaCl, 10 mM BME) and HURP-containing fractions were pooled, concentrated, and buffer exchanged to SEC buffer (20 mM HEPES pH 7.5, 300 mM KCl, 1 mM MgCl$_2$, 1 mM TCEP). The concentrated sample was loaded onto a Superdex 200 10/300 GL column and eluted with 1.2 CV of SEC buffer. Samples containing the protein of interest were pooled, concentrated, aliquoted, and flash-frozen in liquid nitrogen for storage at −80 °C.

HURP$^{1-846}$-eGFP was purified as previously described[12]. Briefly, after cell lysis and centrifugation, the cleared lysate was filtered through a 0.2 μM filter and loaded onto a 5 mL HiTrap SP HP column. After washing, the protein was eluted with a salt gradient ranging from 240 mM to 1 M NaCl. Protein-containing fractions were concentrated, diluted to lower salt, and loaded onto a 1 mL HiTrap Q HP column. The flowthrough was collected, pooled, and loaded onto a 5 mL HisTrap HP. HURP was eluted with an imidazole gradient and buffer exchanged to the final storage buffer E (50 mM HEPES pH 8.0, 300 mM KCl, 1 mM DTT). The concentrated sample was injected onto a Superdex 200 10/300 GL column. Fractions containing HURP$^{1-846}$-eGFP were pooled, concentrated, aliquoted, and flash-frozen in liquid nitrogen for storage at −80 °C.

Kif18A-SNAP constructs were purified as follows. Cell pellets were resuspended, lysed in Kif18A buffer (25 mM Tris pH 7.5, 300 mM KCl, 5 mM MgCl$_2$, 20 mM imidazole, 0.1% Tween-20, 1 mM ATP, 1 mM EGTA, 1 mM DTT, 5% glycerol), and centrifuged as described for the other proteins. The cleared lysates were incubated with 2 mL Ni-NTA agarose resin and equilibrated in the Kif18A buffer for 2 h. The resin was washed four times with 30 mL Kif18A buffer. After washing, the agarose resin was resuspended in 8 mL Kif18A buffer supplemented with Ulp1 protease. Elution with Ulp1 was carried overnight at 4 °C, after which the protein was completely released from the Ni-NTA beads. Beads were separated from the solution by gentle centrifugation and the solution was concentrated to 800 μL. The SNAP tag was labeled with LD655 by incubating 15 nmoles of LD-655 benzylguanine (Lumidyne) with concentrated protein solution for 5 h at 4 °C. The mixture was then centrifuged to remove any aggregates and injected onto a Superdex 200 10/300 GL gel filtration column. Protein-containing fractions were pooled, concentrated, aliquoted, and frozen before storing them at −80 °C.

Kif5B, MAP7, DCX, and Tau were purified as previously described[44,66].

### Cryo-EM sample preparation

Porcine brain tubulin (Cytoskeleton Cat # T240) was reconstituted to 10 mg/mL in BRB80 buffer (80 mM Pipes, pH 6.9, 1 mM ethylene glycol tetraacetic acid (EGTA), 1 mM MgCl$_2$) with 10% (v/v) glycerol, 1 mM GTP, and 1 mM DTT. 10 μL of the tubulin solution were polymerized at 37 °C for 15 min. 1 μL of 2 mM taxol was added to the polymerizing tubulin and incubated at 37 °C for 10 min; this was followed by a second addition of 1 μL taxol and a further incubation of 30 min. Microtubules were pelleted by centrifugation at 37 °C and 15,000 rcf for 20 min. The supernatant containing free tubulin was discarded and the pelleted microtubules were resuspended in resuspension buffer (BRB80 buffer supplemented with 0.05% NP-40, 1.5 mM MgCl$_2$, 1 mM DTT, and 250 μM taxol). After measuring the tubulin concentration in a CaCl$_2$ depolymerized aliquot, the microtubule solution was diluted to 2 μM in dilution buffer (BRB80 buffer supplemented with 0.05% NP-40, 1.5 mM MgCl$_2$, 1 mM DTT, and 100 μM taxol). Immediately before sample preparation, all microtubule-binding proteins were desalted to cryo buffer (BRB80 buffer supplemented with 0.05% NP-40, 1.5 mM MgCl$_2$, 1 mM DTT) using Zeba Spin desalting columns (Pierce).

To prepare microtubule-HURP$^{1-285}$ samples, 2 μL of 2 μM taxol-microtubules were incubated on a glow-discharged holey carbon cryo-EM grid (QuantiFoil, Cu 300 R 2/1) for 30 s, manually blotted with Whatman filter paper, and 2.5 μL of 30 μM HURP$^{1-285}$ were added to the grid. The grid was transferred to a Vitrobot (Thermo Fisher Scientific) set at 25 °C and 80% humidity, and plunge-frozen in liquid ethane after a 1 min incubation with a blot force of 6 pN and a blot time of 6 s.

Microtubule-HURP$^{1-285}$-eGFP-Kif18A$^{1-373}$-SNAP-LD655 sample preparation followed a similar procedure. 2 μL of 2 μM taxol-microtubules were incubated on a glow-discharged holey carbon cryo-EM grid (QuantiFoil, Au 300 R 1.2/1.3) for 30 s, manually blotted as before, and incubated with 2.5 μL of a mixture with 8 μM HURP$^{1-285}$-eGFP, 8 μM Kif18A$^{1-373}$-SNAP-LD655, and 5 mM AMP-PNP. After incubating for 1 min in the Vitrobot under identical conditions as the previous sample, the grid was plunge-frozen and transferred to liquid nitrogen as described before.

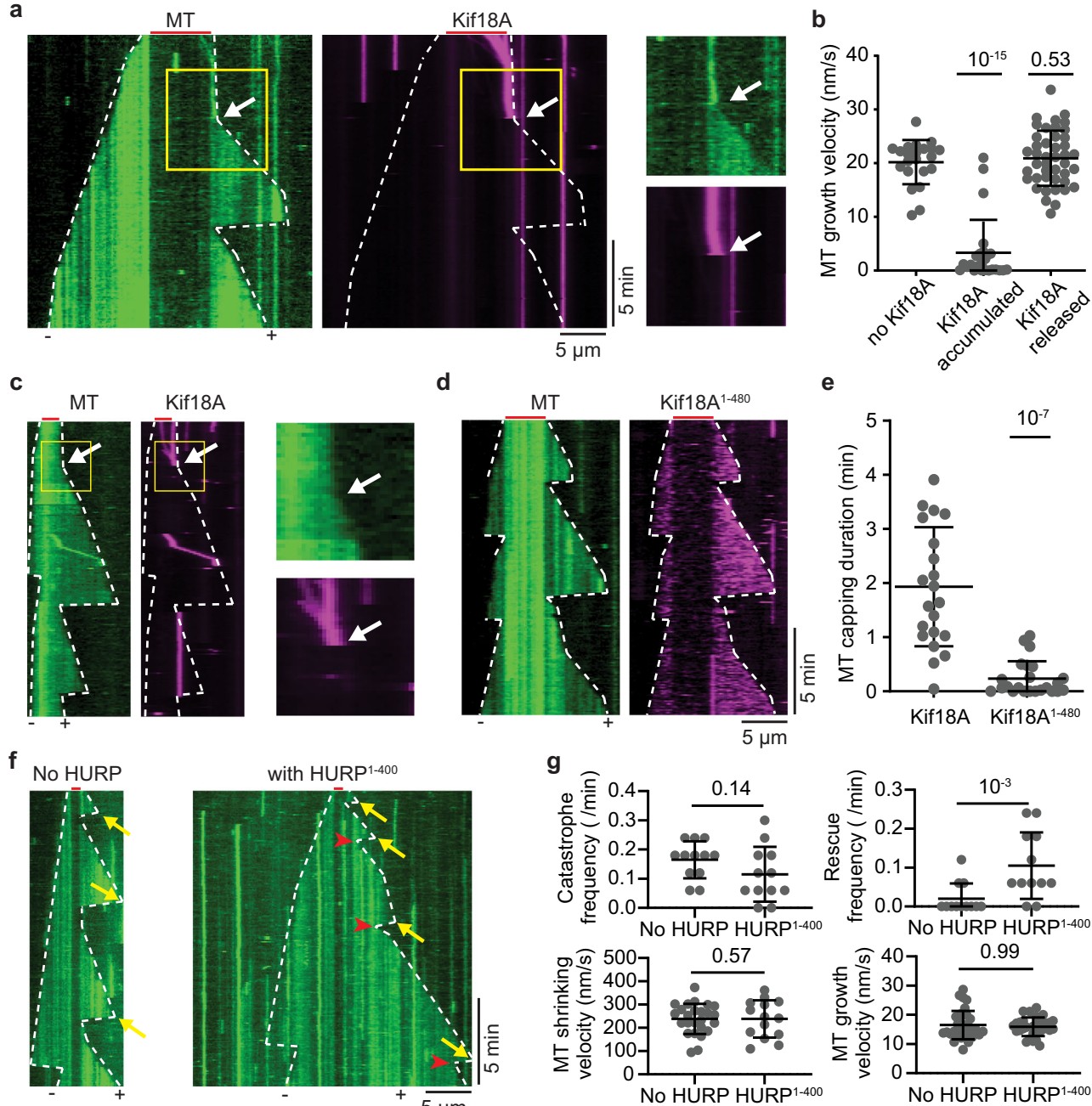

**Fig. 6 | Kif18A acts as a plus-end molecular cap and HURP is a microtubule stabilizer. a** (Left) Kymographs of dynamic microtubules in the presence of 50 nM Kif18A-LD655 preincubated with GMPCPP-microtubule seeds and 2 mM AMP-PNP (no free Kif18A present during imaging). Upon addition of ATP, Kif18A moves on microtubules and accumulates at the plus-end tip. The plus-end of the microtubule stopped growing until the Kif18A accumulation was spontaneously released. (Right) Magnified view of the area inside yellow rectangles. **b** Microtubule plus-end growth velocities with Kif18A accumulated at the plus-end or released from the plus-end (from left to right, $N = 22, 25, 43$ microtubule growth periods). **c** (Left) Kymographs of dynamic microtubules with 50 nM Kif18A-LD655 preincubated on GMPCPP-microtubule seeds and 2 mM AMP-PNP, imaged in the presence of 50 nM free Kif18A-LD655 after exchange to the final imaging buffer. (Right) Magnified view of the area inside yellow rectangles. **d** Kymographs of dynamic microtubules with Kif18A[1-480]-LD655 preincubated with GMPCPP-microtubule seeds and 2 mM AMP-PNP, imaged in the presence of 50 nM free Kif18A[1-480]-LD655. **e** Duration of the

microtubule plus-end capping by Kif18A or Kif18A[1-480] (from left to right, $N = 21, 23$ kymographs). **f** Kymographs of dynamic microtubules with or without 500 nM HURP[1-400]-eGFP. Yellow arrows represent catastrophe events and red arrowheads represent rescue events. **g** Catastrophe and rescue frequencies (top; $N = 12$ kymographs for each condition), microtubule plus-end shrinking velocities (lower left; $N = 25, 14$ microtubule shrinking periods from left to right), and microtubule plus-end growth velocities (lower right; $N = 34, 31$ microtubule growth periods from left to right) with or without 500 nM HURP[1-400]-eGFP. In (**a**, **c**, **d**, **f**), white dashed lines show the track of microtubule ends and red lines mark the position of the seeds. In (**a**, **c**), white arrows mark the release event. In (**b**, **e**, **g**), the center line and whiskers represent the mean and S.D., respectively. $P$ values are calculated from a two-tailed $t$-test. The proteins were flown together with unpolymerized tubulin before imaging. Each imaging duration is 1000 s. The in vitro motility assays were performed with 3 technical replicates. Source data are provided as a Source Data file.

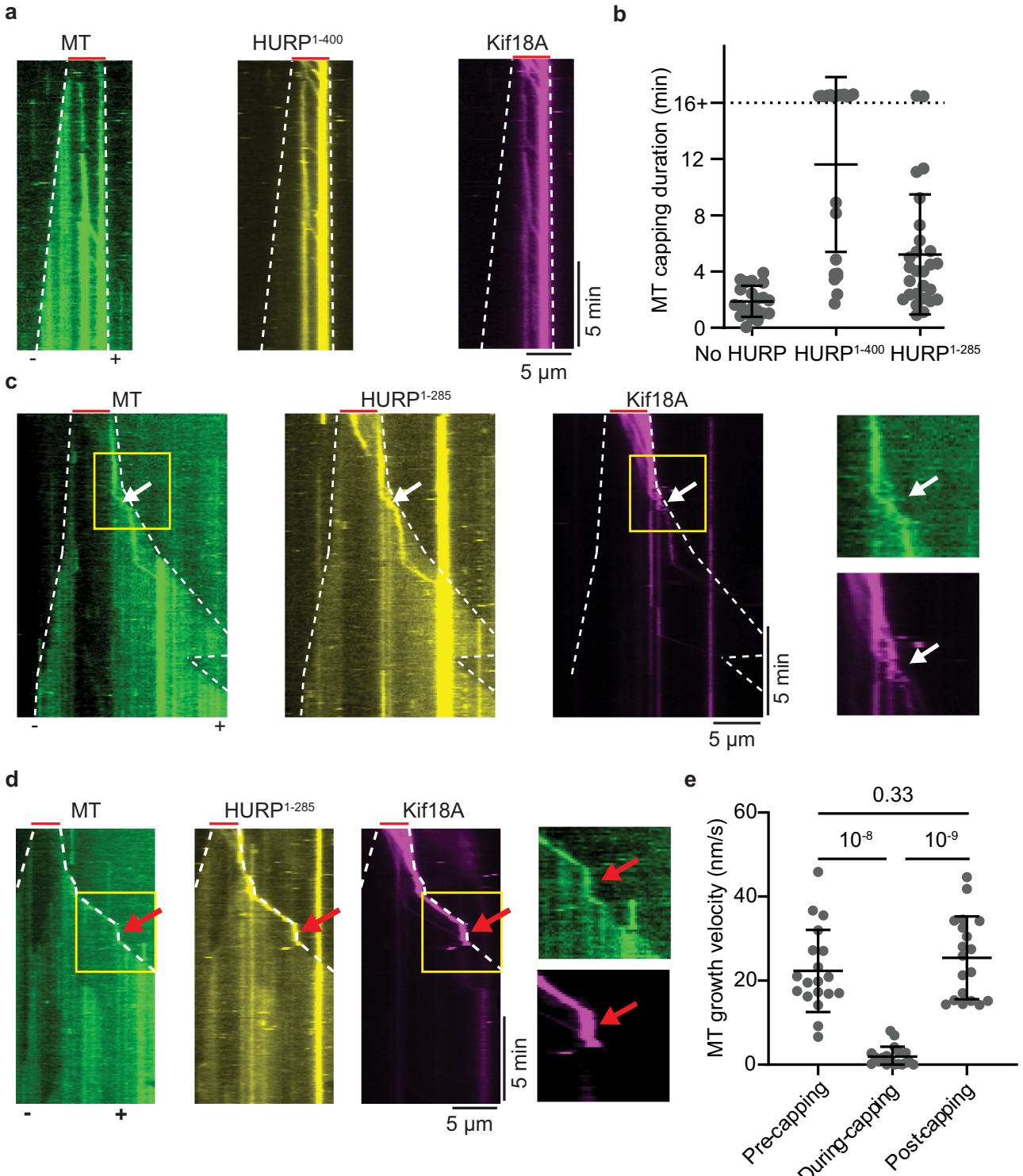

**Fig. 7 | Kif18A and HURP synergistically suppress microtubule dynamics.**
**a** Kymographs of dynamic microtubules in the presence of 50 nM Kif18A-LD655, 500 nM HURP$^{1-400}$-eGFP, and 2 mM AMP-PNP. **b** Duration of the microtubule plus-end capping by Kif18A in conditions of no HURP, 500 nM HURP$^{1-400}$-eGFP or 500 nM HURP$^{1-285}$-eGFP (from left to right, $n$ = 21, 22 and 27 kymographs). **c** Kymographs of dynamic microtubules in the presence of 50 nM Kif18A-LD655, 500 nM HURP$^{1-285}$-eGFP, and 2 mM AMP-PNP. Capping is observed from $t$ = 0 s. White arrows mark the release event ($N$ = 19 capping events for each condition). **d** Representative kymographs for identical conditions as in (**c**) with capping following growth. Red arrows mark the presence of the cap after initial growth. **e** Microtubule growth velocities

before, during, and after capping ($N$ = 19 capping events for each condition). In (**a, c**, and **d**), white dashed lines show the track of microtubule ends and red lines mark the position of the GMPCPP seeds. In (**c, d**), yellow rectangles highlight the area of the kymograph where the motor release occurs. In (**b, e**), the center line and whiskers represent the mean and S.D., respectively. $P$ values are calculated from a two-tailed $t$-test. The proteins were pre-mixed and flown together with unpolymerized tubulin before imaging. Each imaging duration is 1000 s. The in vitro motility assays were performed with 3 technical replicates. Source data are provided as a Source Data file.

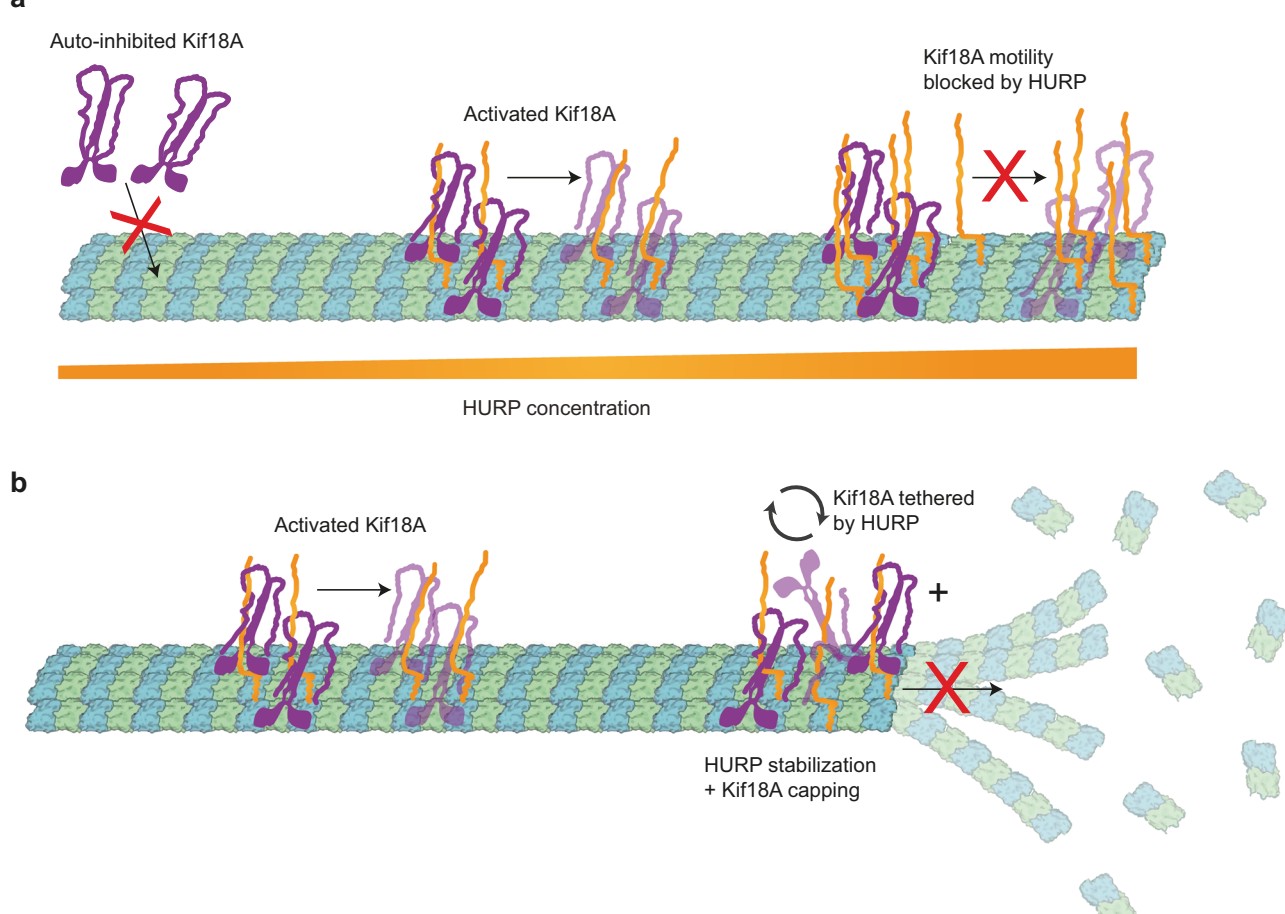

**Fig. 8 | Model of HURP and Kif18A interplay in motility and microtubule dynamics. a** Schematic representation of the concentration-dependent regulation of Kif18A motility by HURP[44]. Microtubule-bound HURP helps recruit Kif18A and releases its auto-inhibition. However, excess HURP decreases Kif18A run time and velocity. **b** Schematic representation of microtubule dynamics regulation at the plus-end by HURP and Kif18A. The presence of HURP and Kif18A limits both growth and catastrophe of the microtubule plus-end, regulating microtubule length.

## Cryo-EM data collection

Data for microtubules decorated with HURP[1-285] and HURP[1-285] eGFP-Kif18A[1-373]-SNAP_LD655 were collected using an Arctica microscope (Thermo Fisher Scientific), operated at an accelerating voltage of 200 kV (Supplementary Table 1). All cryo-EM images were acquired on a K3 direct electron detector (Gatan), at a nominal magnification of 36,000×, corresponding to a calibrated physical pixel size of 1.14 Å. The camera was operated in superresolution mode, with a dose rate of ~7.2 electrons/pixel/s on the detector. We used an exposure time of ~9 s dose-fractionated into 50 frames, corresponding to a total dose of ~50 electrons/Å² on the specimen. All the data were collected semi-automatically with the SerialEM software package[67].

## Cryo-EM image processing

For the microtubule-HURP[1-285] dataset, the movie stacks were imported to CryoSparc and motion-corrected[68]. The CTF parameters were estimated with the patch CTF job and manually curated to remove bad micrographs. Particles were automatically picked with the filament tracer, initially without a reference, and later using 2D templates as input. The segment separation was set to 82 Å, corresponding to the length of an α-β tubulin dimer. Particle images were extracted with a box size of 512 pixels and Fourier-cropped to 256 pixels for initial image processing. These images were subjected to 2 rounds of 2D classification and classes showing clear density for the microtubule were selected; classes showing blurry density, junk particles, or non-centered microtubules were discarded. Microtubules with different numbers of protofilaments were separated through a heterogeneous refinement job where 13 and 14 PF lowpass-filtered references were used as initial models. In both cryo-EM datasets, the 14 PF population corresponded to the majority class, and therefore was selected for further processing. 14 PF particles were subjected to a helical refinement with an initial rise estimate of 82.5 Å and a twist of 0°. Angular assignments and shifts were further refined through a local refinement using a hollow cylindrical mask enclosing the microtubule reconstruction. To obtain a reconstruction that accounts for the presence of a symmetry-breaking seam on the microtubule, we used a Frealign-based seam search routine with custom scripts that determine the seam position on a per-particle basis[34]. For this purpose, we converted the CryoSparc alignment file from the last local refinement, first to the STAR format using the csparc2star script from the PyEM suite (10.5281/zenodo.3576630), and then to PAR Frealign format using a custom Python script. Upon completion of the seam search protocol, the particles were imported back to CryoSparc with the seam-corrected alignments and re-extracted without Fourier cropping (box size: 512 pixels), using the new improved alignments to recenter the picks before extraction. A volume was reconstructed with the imported particles through a local refinement job and a local CTF refinement was performed to estimate the per-particle CTF. Another local refinement was run on this particle set to produce the final C1 reconstruction. The symmetry search job was used to determine the rise and the twist of

the map, and these parameters were input to a symmetry expansion job to fully exploit the pseudo-symmetry of each microtubule particle. A local refinement was performed on the newly expanded particle stack with a cylindrical mask around the microtubule to yield the final symmetrized reconstruction of the entire microtubule. A final local refinement was performed with a smaller mask around the good PF (PF opposite to the seam) and the PF adjacent to it, producing a 3.1 Å resolution map of HURP bound to microtubules (Supplementary Figs. 6 and 7a).

The microtubule-HURP$^{1-285}$-Kif18A$^{1-373}$ dataset was processed in an identical way up to the step where the particle set is symmetry expanded and the final symmetrized reconstruction of the entire microtubule is generated. From this point, we generated a new local refinement with a shaped mask encompassing 2 tubulin dimers, 2 Kif18A monomers, and the inter-PF groove where HURP inserts. This produced a consensus reconstruction at 2.9 Å resolution that was further sorted using 3D classification, with a more constrained map only covering single copies of a tubulin dimer, a Kif18A, and HURP molecule. Initially, a random subset containing 10% of the refined particles was used for reference-free 3D classification without alignment, using the consensus reconstruction and aligned particles as inputs. This generated 5 classes that were used as initial references for a second alignment-free 3D classification job on the full particle stack. Some of these classes contained density for tubulin + HURP, others showed tubulin + Kif18A, and one class only showed density for tubulin. The second 3D classification job recapitulated the results from the first classification and allowed us to computationally sort the compositional heterogeneity present in the consensus reconstruction. Classes showing no distinct features with each other were combined and locally refined, yielding a 3.5 Å resolution class for tubulin, a 3.1 Å resolution map for tubulin + Kif18A, and a 3.0 Å resolution reconstruction for tubulin + HURP (Supplementary Figs. 7b and 12). Cryo-EM processing parameters are shown in Supplementary Table 1.

### Model building and refinement

The final cryo-EM map from the microtubule-HURP$^{1-285}$ dataset was used for modeling of HURP MTBD1. Tubulin dimers from a previous publication (PDB: 6DPV)[69] were rigid-body fitted into the density map using ChimeraX[70]. The local resolution of the map region corresponding to HURP and the presence of 3 sequential bulky side-chains in the inter-PF groove HURP density (R122, Y123, R124) allowed us to unambiguously assign the protein sequence register. HURP modeling was performed in Coot[71] by manually tracing the main chain and assigning the corresponding residues. In the case of the α-helical density, it was modeled as a perfect α-helix and the register was identified by the presence of bulky side chains L94 and Y97. The HURP and tubulin models (2 tubulin dimers, 1 HURP) were then combined in a single PDB file and real-spaced refined using the Phenix software[72]. For microtubule-Kif18A modeling a similar procedure was followed starting from PDB ID 5OCU[36], manually applying changes and then refining it against the Kif18A-containing density map. Model refinement statistics are shown in Supplementary Table 1. For visualization purposes and figure generation, refined models for α-β-tubulin, HURP, or Kif18A were superimposed with the corresponding maps, colored based on subunit type (α-tubulin in green, β-tubulin in blue, HURP in orange, and Kif18A in purple), and the maps were colored and segmented accordingly, creating independent maps for the four subunit types. The splitted maps for the different types of subunits were set to different thresholds that better reflected their average local resolution. Lower and identical threshold values were used for α and β tubulins to highlight their higher resolution features, while the thresholds for HURP and Kif18A were set independently.

Lattice compaction, derived from inter-dimer and intra-dimer distances, was obtained from atomic models representing either HURP-bound taxol microtubules or free taxol microtubules by averaging the Cα distances between α- and β-tubulin for corresponding residue pairs. The atomic model for the taxol microtubule state was obtained by refining atomic coordinates in a cryo-EM map of taxol-stabilized microtubules (3.3 Å), in the absence of any microtubule-associated protein.

### Structure prediction

Human HURP$^{285-400}$ and Kif18A$^{1-480}$ sequences were used for AlphaFold 3[32] structural prediction in the web server (https://alphafoldserver.com/) to generate five models and their associated statistics and confidence metrics (for plDDT scores and PAE matrices, see Supplementary Fig. 5).

### Native tubulin extraction

We extracted native tubulin from pig brain through a series of polymerization and depolymerization cycles. Initially, the pig brain tissue was lysed and then mixed in a 1:1 mass-to-volume ratio with a depolymerization buffer (50 mM MES, 1 mM CaCl$_2$, pH 6.6 with NaOH). This mixture was centrifuged at 32,000 rcf and 4 °C. The supernatant was then combined with high molarity PIPES buffer (HMPB) (1 M PIPES free acid, 10 mM MgCl$_2$, and 20 mM EGTA, pH 6.9 with KOH) and glycerol in a 1:1:1 volume ratio. Subsequently, GTP and ATP were added to reach final concentrations of 0.5 mM and 1.5 mM, respectively. This mixture was incubated at 37 °C for 1 h, followed by centrifugation at 400,000 rcf for 30 min at 37 °C.

After this process, the pellet was resuspended in the depolymerization buffer and incubated at 4 °C for 15 min to induce depolymerization. The supernatant obtained was then mixed again with HMPB and glycerol in a 1:1:1 volume ratio, with the addition of GTP and ATP to the previously stated concentrations. The mixture underwent another incubation at 37 °C for 1 h, followed by centrifugation at 400,000 rcf for 30 min at 37 °C. The final pellet was resuspended in cold BRB80 buffer and incubated at 4 °C for 30 min. A last centrifugation at 400,000 rcf for an unspecified duration at 4 °C was performed, and tubulin was diluted to 34 mg/mL before the tubulin was stored at −80 °C.

### Biotin or Cy3 labeling of tubulin

To label tubulin with biotin or Cy3, NHS ester labeling was performed on polymerized microtubules. Initially, GTP and DTT were added to 0.2 mL of tubulin aliquots, adjusting the concentrations to 5 mM each. This mixture was incubated at 37 °C for 30 min to promote microtubule polymerization. Following polymerization, the mixture was centrifuged at 400,000 rcf at 37 °C for 30 min, overlaying it with 0.5 mL of warm high pH cushion (0.1 M HEPES, pH 8.6, 1 mM MgCl$_2$, 1 mM EGTA, 60% (v/v) glycerol) to enhance pellet separation. The supernatant above the cushion was then discarded, and the interface between the tube wall and cushion was gently washed twice with 250 μL of warm labeling buffer (0.1 M HEPES, pH 8.6, 1 mM MgCl$_2$, 1 mM EGTA, 40% (v/v) glycerol) before complete removal of the supernatant and cushion. Subsequently, the pellet was resuspended in 0.4 mL of warm labeling buffer, to which 50 μL of 6 mM NHS-biotin or NHS-Cy3 in DMSO was added. This solution was then incubated at 37 °C for 30 min on a roller mixer.

To eliminate free NHS-biotin or NHS-Cy3, the microtubule mixture underwent centrifugation at 400,000 rcf at 37 °C for 30 min using a low pH cushion (BRB80 with 60% (v/v) glycerol). The supernatant above the cushion was removed, and the tube wall-cushion interface was rinsed twice with 250 μL of warm BRB80 before discarding the supernatant and cushion. The tube wall was then washed again twice with 250 μL of warm BRB80. The pellet was resuspended in cold BRB80 and incubated at 4 °C for 30 min to allow microtubule depolymerization. Finally, the mixture was centrifuged at 400,000 rcf at 4 °C for 30 min, and the supernatant was collected and stored in a −80 °C freezer.

## Preparation of GMPCPP-microtubule seeds

GMPCPP-microtubule seeds were used for the dynamic microtubule assays. The ultracentrifuge, rotor, tubes, and BRB80 buffer (80 mM PIPES (Free Acids), 1 mM MgCl$_2$, 1 mM EGTA, 1 mM DTT, pH adjusted to 6.8 with KOH, ensuring a pH below 7) with 10% DMSO were pre-cooled. A mixture was then made with unlabeled tubulin, 5% biotin-tubulin, and 5% Cy3-labeled tubulin (Cy3-labeled tubulin is optional). The mixture was diluted to 1–3 mg/mL with cold BRB80 containing 10% DMSO and incubated on ice for 10 min (for longer microtubules, a concentration of 0.3–0.5 mg/mL was used). The mixture was cold spun at 400,000 rcf for 10 min to remove inactive tubulin, and the supernatant was collected, to which GMPCPP was added to achieve a final concentration of 1 mM before incubation at 37 °C for 20 min (extended to 90–120 min for longer microtubules). After warming the centrifuge equipment and buffer to 37 °C, the tubulin mix was spun at 37 °C at 400,000 rcf for 10 min. The supernatant was discarded, and the pellet was gently resuspended in 25–50 μL of warm BRB80 buffer (using a cut pipette tip to avoid shearing of the microtubules). The GMPCPP microtubule seeds were kept at room temperature in the dark and used within 2 weeks.

## Preparation of taxol-stabilized microtubules for TIRF

Taxol-stabilized microtubules were used for single-molecule motility imaging. They were made by diluting 4 μL of 34 mg/mL of tubulin, 5% of which was biotin-labeled and 5% which was Cy3-labeled, into 46 μL BRB80 (80 mM PIPES at pH 6.8, 1 mM MgCl$_2$, and 1 mM EGTA). This mixture was then added to an equal volume of polymerization solution (1X BRB80 with 2 mM GTP and 20% DMSO). The tubulin was incubated at 37 °C for polymerization for 40 min, after which 10 nM of taxol was added and the mixture was incubated for another 40 min. The microtubules were pelleted by centrifugation at 20,000 rcf for 15 min at 37 °C and then resuspended in 25 μL BRB80 solution with 10 nM taxol and 1 mM DTT. The taxol-stabilized microtubules were kept at room temperature in the dark and used within 2 weeks.

## Fluorescence microscopy

Fluorescence imaging utilized a custom-built multicolor objective-type TIRF microscope, incorporating a Nikon Ti-E microscope body, a 100× magnification 1.49 N.A. apochromatic oil-immersion objective (Nikon), and a Perfect Focus System. Detection of fluorescence employed an electron-multiplied charge-coupled device camera (Andor, Ixon EM+, 512 × 512 pixels), with an effective camera pixel size of 160 nm post-magnification. Excitation of GFP, Cy3, and LD655 probes occurred via 488 nm, 561 nm, and 633 nm laser beams (Coherent) delivered through a single mode fiber (Oz Optics), with emission filtering accomplished using 525/40, 585/40, and 697/75 bandpass filters (Semrock), respectively. Microscope operations were managed via MicroManager 1.4.

## Preparation of flow chambers

Glass coverslips were coated with polyethylene glycol (PEG) to reduce nonspecific protein binding. First, plain glass coverslips underwent sequential cleaning steps involving water, acetone, and water sonication for 10 min each, followed by a 40 min sonication in 1 M KOH using a bath sonicator.

A "piranha" cleaning step was performed for additional cleaning of the glass: piranha solution preparation and usage were carried out in a fume hood with appropriate PPE, including heavy-duty chemical gloves, face shield, chemical apron, and sleeves. A 500 mL beaker was placed on ice. 75 mL of 30% H$_2$O$_2$ was scaled in a 125 mL flask and poured into the beaker. 125 mL of concentrated H$_2$SO$_4$ was scaled in a 250 mL flask and slowly added to the H$_2$O$_2$ while shaking the beaker. Samples were sonicated for 45 min in this solution. Subsequently, the coverslips were rinsed 4× with water and 3× with methanol, immersed in 3-Aminopropyltriethoxysilane in acetate and methanol for 10 min with 1-min sonication intervals between steps, further cleaned with

methanol, and air-dried. A 30 μl volume of 25% biotin-PEG-succinimidyl valerate in a NaHCO$_3$ buffer (pH 7.4) was applied between two coverslip pieces and left to incubate at 4 °C overnight. Following incubation, the coverslips were cleaned with water, air-dried, vacuum sealed, and long-term stored at −20 °C. Flow chambers were constructed by sandwiching double-sided tape between a PEG-coated coverslip and a glass slide.

## Single-molecule motility imaging

The flow chambers underwent a 2-min incubation with 5 mg/ml streptavidin, followed by washing with MB buffer (composed of 30 mM HEPES pH 7.0, 5 mM MgSO$_4$, 1 mM EGTA, 1 mg/ml casein, 0.5% pluronic acid, 0.5 mM DTT, and 1 μM taxol). Subsequently, the chamber was incubated with biotinylated microtubules for 2 min and washed again with MB buffer. Proteins were then diluted to desired concentrations in imaging buffer (MB buffer supplemented with 150 mM KAc, 0.15 mg/ml glucose oxidase, 0.025 mg/ml catalase, 0.8% D-glucose, and 2 mM ATP), and introduced into the flow chamber. Motility was recorded over a 2-min period. For the HURP dragging assay, 300 nM of Kif18A and 25 nM HURP were mixed into imaging buffer and flowed into a chamber with immobilized taxol microtubules. The single-molecule motility imaging was performed at 22 °C.

Run frequencies were calculated by counting the number of running molecules on each kymograph and dividing them by the length of the microtubule and the duration of imaging. Run frequencies were normalized to the first dataset of each plot.

## Dynamic microtubule imaging

To conduct a dynamic microtubule assay, tubulin, HURP, and Kif18A were cold centrifuged at 400,000 rcf for 10 min to eliminate protein aggregates. Subsequently, biotinylated GMPCPP-microtubule seeds (with or without Cy3 labeling) were incubated on a biotin-PEG glass surface via Streptavidin bonding. The chamber was then flowed with a dynamic microtubule mixture, (1X BRB80 pH 6.8, 1 mg/ml casein, 0.5% pluronic acid, 0.5 mM DTT, 75 mM KAc, 2 mg/mL unlabeled tubulin, 0.05 mg/mL Cy3-tubulin, 4 mM GTP, 1 mM ATP, and 0.2% methylcellulose), along with the desired concentrations of Kif18A and/or HURP. The dynamic microtubule imaging was performed at 22 °C.

For the pre-incubation of Kif18A on the microtubule seeds, Kif18A was pre-incubated in MB buffer containing 150 mM KAc and 2 mM AMP-PNP for 10 min to enhance its affinity to microtubules.

## HURP binding curve fitting

HURP binding intensity was calculated in MATLAB 2023a by averaging data from individual microtubules. Curve fitting was performed using Origin 8.5. Binding curves of fluorescently labeled HURP were fit to the Hill equation: $I_{HURP} = \frac{[HURP]^n}{[HURP]^n + (K_d)^n}$, where $I$ represents the normalized fluorescence intensity of HURP on the microtubules, $K_d$ is the half-maximal saturation concentration, and $n$ is the Hill coefficient.

## Statistics and reproducibility

Single-molecule motility experiments were conducted with a minimum of two independent technical replicates to ensure reproducibility. Data points were randomly selected without exclusion, and no statistical methods were applied to predetermine sample size.

## Reporting summary

Further information on research design is available in the Nature Portfolio Reporting Summary linked to this article.

# Data availability

Materials can be obtained from A.Y. and E.N. under a material transfer agreement with the University of California, Berkeley. The structural coordinates for HURP and Kif18A in complex with tubulin have been

deposited in the Protein Data Bank (PDB) under accession codes 9DHZ and 9DIO, respectively. Coordinates of the tubulin models used for determining lattice compaction have also been deposited in the PDB under accession codes 9DXC for the HURP-taxol microtubule, and 9DXE for the taxol microtubule. Additionally, the cryo-EM maps from this study are available in the Electron Microscopy Data Bank (EMDB) under accession codes EMD-46893 (HURP-taxol microtubule), EMD-46894 (Kif18A-taxol microtubule) and EMD-47285 (taxol microtubule). Source data are provided with this paper.

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

## Acknowledgements

We thank J. Peukes and Z. Yang for helpful discussions related to image processing and biochemical design, L. Wordeman for Kif18A plasmid Addgene deposition, M. Esbin for supplying the U2OS cells used in cloning experiments, D. Toso, R. Thakkar, P. Tobias, K. Stine, and V. Marquez for their support with cryo-EM data collection and computation infrastructure, the UCSF ChimeraX team for the software development used in structural rendering, V. Perez-Bertoldi for assisting with graphic design and illustration, Y. He for producing the pegylated glass surface used in TIRF microscopy and J. Fernandes for tubulin preparation. Work was funded by NIGMS (R35GM127018 to E.N., R35GM136414 to A.Y.), by the European Research Council (ERC-2022-SYG to E.N.), and by NSF (MCB-1055017, MCB-1617028 to A.Y.). E.N. is a Howard Hughes Medical Institute Investigator.

## Author contributions

J.M.P.B., A.T., and E.N. initially conceived the project and all authors contributed to its development and the design of experiments. J.M.P.B. purified proteins, performed cryo-EM and single-molecule TIRF experiments, analyzed data, performed modeling and in silico predictions. Y.Z. performed single-molecule TIRF experiments, dynamic microtubule imaging, and analyzed data. E.N. and A.Y. supervised the project and secured funding. J.M.P.B., Y.Z., E.N., and A.Y. wrote the manuscript, with further edits by all authors.

## Competing interests

The authors declare no competing interests.
