## [Transparent Peer Review file · Nature Communications]

HURP regulates Kif18A recruitment and activity to synergistically control microtubule dynamics

Corresponding Author: Professor Eva Nogales

Version 0:

Reviewer comments:

Reviewer #1

(Remarks to the Author)

Kinesin-8 are dual activity kinesins with both motile and microtubule depolymerization properties. They are important for the regulation of microtubule dynamics and spindle organization, and their activity needs to be regulated. During division in human cells, the kinetochore fibers recruit multiple proteins including the kinesin-8 Kif18A and Hepatoma upregulated protein (HURP) which is thought to modulate Kif18A activity. In this manuscript, the authors describe the complex regulation of human Kif18A by HURP. The demonstration includes a solid series of single molecule experiments performed on a range of constructs of both Kif18A and HURP. The single molecule work is complemented by a high quality cryo-EM analysis of the microtubule interaction with HURP alone or together with a monomeric Kif18A construct in AMPPNP. The single molecule experiments show a non-linear effect of HURP on Kif18A activity. The authors manage to rationalize mechanistically most of the single molecule results using the cryo-EM results, and provide a high-resolution structure of HURP interacting with a microtubule. The conclusions are solid, the manuscript is well written and well polished. Overall, this is an important contribution to the regulation of molecular motors and I'm very enthusiastic about this paper being published in Nature Communications once the comments below are addressed.

Note: In the document I received the line numbers are not unique: there is for instance a line 668 in the middle of both pages 34 and 35, so the comments below are given with both a page number and a line number.

Main comments:

1. On the representative images shown in Figure 1C for HURP, a substantial amount of fluorescence appears to originate from clusters (/aggregates/droplets). These clusters are possibly more prevalent in the images selected for the two longest constructs. Does Kif18A interact with these HURP clusters? Is it possible to rule out a scenario in which the longer the HURP construct the more HURP protein segregates into these clusters (to the expense of the HURP fraction that binds microtubule)? Such a scenario could for instance contribute to the trend seen in panel 1F – normalized run frequency (the longer the HURP construct the less Kif18A is activated/recruited).

2. page 6 - line 134-135 "No density corresponding to the HURP MTBD2 was present in the map". This statement should ideally be supported by showing in a supplementary figure a ~8-12 Å low passed filtered version of the reconstruction without any mask applied nor sharpening and displayed at low threshold (or something equivalent).

3. page 7-9 - lines 154-187. This part would benefit from a brief comment on the kinesin motor state as explained here. The reported Kif18A structure resolution (in the 3-4 Å range) is a considerable improvement over the previous microtubule bound Kif18A structures (reported to be around 5 Å ; Locke, J. et al, PNAS, 2017, PMID:29078367). The resolution obtained in the present work is similar to the one reported with a yeast kinesin-8 Kip3 (Hunter, B. et al, Nat Commun., 2022, PMID:35859148) obtained with a similar processing strategy. It would be informative to compare the conformation of the AMPPNP-bound Kif18A motor obtained by the authors with the ones of the latter paper, which reported in AMPPNP a motor conformation with a nucleotide binding pocket not closed and without the neck-linker docked. This way the reader would know with which conformation of this monomeric Kif18A motor (e.g. open/closed/semi-open) the cryo-EM experiment with Kif18A+HURP was done (since one can't strictly rule out that a different HURP behavior could occur with a different kinesin motor conformation). Secondly, this comment on the Kif18A motor state would also shed some light on Kif18A mechanochemistry and be of interest in the kinesin field.

4. The authors should explicit if the different single molecule traces used in a given plot originate from a single experiment (e.g. same protein prep and collected on the same microscope session) or from independent experiments, and in that case where the independence between experiments lies.

5. Clarification of some physico-chemical conditions:

- page 6 - lines 128-129 and page 32 - lines 644-645. The expression "in the absence of added salt" is ambiguous and should be replaced by an expression that explicit the physico-chemical condition of the interaction (e.g. in buffer X at x°C).
- I have not seen any information regarding the temperature(s) used in the single molecule experiments. This/These temperatures should be stated (at least) in the material and methods ("Single-molecule motility imaging" and "Dynamic microtubule imaging", on page 25)

6. page 23 - line 519-520: "The GMPCPP microtubule seeds were usually good for 2 weeks." & page 24 - line 528-529 "The microtubule is stocked at room temperature and good to use for 2 weeks."

Although these statements maybe useful to the readers, what is "good" is not defined and as importantly the statements do no clearly report what was done time wise with these seeds/microtubule. It would be preferable to state more what was done so that the reader can rule out any bias that could originate from variations in the materials (e.g. should different concentration conditions in a plot originate from fresh microtubules and others from 2 weeks old microtubules).

7. page 28 - line 614: "Kd is determined from a fit to binding isotherm" and page 32 - line 646-647: "The fit used to determine the Kd is determined from a fit to binding isotherm"

These informative fits - which are not mentioned in the material and methods - likely assume that each HURP molecule binds the microtubule independently (i.e. with no cooperativity). Did the authors noticed a significant difference in the helical parameters of the microtubule decorated with HURP compared to a naked microtubule of the same type stabilized similarly ?

Minor comments:

page 3 - line 68: "to visualize how HURP and Kif18A bind to the microtubules at near-atomic resolution." "near-atomic resolution" being not well defined, it could be replaced by the actual resolution range of the complex.

page 16 - line 365: "minutes. microtubules were": capital letter needed for microtubules.

page 18 - line 401: "the length of an α - β tubulin dimer": the beta symbol doesn't have the same font as the other alpha/beta symbols.

page 24 - line 541: "Glass coverslips were coated s with polyethylene": I think there is a free "s" after coated.

page 25 - line 570: "containing 150 mM KAc and AMPPNP": the concentration of AMPPNP was omitted.

page 28 - line 614: "S.D." and page 30 line 637: "S.E.". These common abbreviations should likely be explicit the first time they are used in the text.

page 33 - Fig 4 E. The relative orientation of the class averages could be indicated for non specialists (like what was done in panel D of that same figure).

page 34 - line 668: "monoKif18A" is not defined (presumably Kif18A 1-373 / monomeric Kif18A).

page 42: The FSC curves plots in Supplementary Figure 5 are very small and hard to read. They should be scaled up.

page 45 - line 753: "microtubule+HURP+Kif18A" dataset: use of "+" instead of "-" as in most of the paper, for instance in the line just above (752), this is written: "microtubule-HURP-Kif18A dataset". This occurs in several places and the use of "-" or "+" could be homogenized throughout the text.

page 50: The headers of the supplementary table 1 should explicit which specific constructs of HURP and Kif18A were used.

Reviewer #2

(Remarks to the Author)

In the present manuscript, Perez-Bertoldi and co-workers investigate the molecular mechanism underlying the regulation of the kinesin-8 Kif18a by the Ran-GTP regulated MAP HURP. For some years now, these two proteins have a conspicuous enrichment at the chromosome proximal ends of kinetochore microtubules (kMTs) and have been implicated in chromosome congression by regulating kinetochore microtubule dynamics. However, the mechanism remained unclear, despite previous compelling in vivo evidence (mostly from Ye, F. et al. HURP Regulates Chromosome Congression by Modulating Kinesin Kif18A Function. *Curr.Biol.* 21 , 1584–1591 (2011)), showing that HURP overexpression inhibits Kif18a activity. By resourcing to in vitro reconstitution analyses by TIRF microscopy and Cryo-EM studies, the authors now provide a near-atomic resolution explanation for how HURP regulates Kif18a activity. The key finding was that, at high concentration, HURP competes with Kif18a for tubulin binding causing a steric clash. Interestingly, low HURP concentrations seem to activate Kif18a motility (through a HURP N-terminal activation domain), but this scenario seems unlikely in the context of their enrichment on kMTs and regulation of chromosome oscillations. The authors also report on possible Kif18a auto-inhibitory activity by its own C-terminal tail, similar to other kinesins. Another clarifying aspect is related with the controversial mechanism by which Kif18 regulates microtubule dynamics. Clearly, Kif18 is able to depolymerize stable

microtubules (in the absence of free tubulin) on its own, but it does so rather inefficiently and in a manner that can be suppressed by HURP, supporting a microtubule capping model where Kif18a, assisted by HURP, suppresses microtubule plus-end dynamics to control kMT length. Overall, this is a technically very sound paper and the data are robust to sustain the proposed mechanism. I am therefore strongly supportive of publication of the present work and have only very minor textual suggestions, as detailed below:

1- Title: I find it misleading to include "mitotic spindle regulation" in the title of a paper that has no single experiment on mitotic spindles. I would recommend replacing by "microtubule length control".

2- Introduction, page 2, line 52: it is yet unclear what the role of chromosome oscillations is and whether it serves any purpose in the alignment of chromosomes during mitosis. Without getting into detail on k-fiber length-dependent models of chromosome congression, a naïve reader might actually perceive that oscillations move chromosomes away from the equator, thus opposing chromosome alignment. I would either dwell a bit on models of chromosome congression or re-write the sentence for simplicity.

3- Results, page 6: Could the authors comment on the KD differences found for HURP binding to microtubules with or without physiological salt and discuss possible scenarios in which HURP might bind to microtubules without saturation (closer to physiological conditions) and respective implications for Kif18a activity/function?

4- Discussion, page 12, line 272: yeast actually do not have a HURP orthologue. Thus, the "antenna model" might still hold true in yeast or even play a synergistic role with HURP in systems where HURP is present. This should be clarified.

Reviewer #3

(Remarks to the Author)

Perez-Bertoldi and Zhao used state-of-the-art single-molecule and structural approaches to address the dynamics of Kif18a and HURP on microtubules. How microtubule-associated-proteins control the movement of motor proteins on microtubules is an interesting and important problem. Here, the authors focus on the interplay between HURP and Kif18a. Key advances from the paper are high-resolution structures of HURP and Kif18a on MTs and the concentration-dependent effects of HURP on Kif18a-microtubule interactions. At low concentrations, HURP promotes Kif18a motility. Probably due a release of Kif18a autoinhibition. At high HURP concentration the Kif18a motility is impaired. Probably due their MT binding in a competitive manner.

This is an interesting mode of action. Suppressed dynamics at the MT plus-end in the presence of HURP and Kif18a provides a clue for the regulation of spindle dynamics in vivo. Although implications of the proposed model on spindle length and k-fiber stability are not tested in cells, the in vitro experiments provide quantitative insights into HURP and Kif18a on microtubules under well-controlled conditions.

The authors addressed HURP-Kif18a / MAP-motor interplay on microtubules from a structural and single-molecule perspective, resulting in new insights that are of broad interest to the microtubule community. I therefore support the publication of this work and encourage the authors to address the following points:

Major points

1)
The authors speculate that the synergistic effects of HURP and Kif18a on stabilizing dynamics at the MT plus-end reflects an interaction between HURP (285-400) and Kif18a (1-480). This explains data in Figures 1 and 2 of the paper and is consistent with Figure 5 and Suppl. Figure 13. Direct evidence for this interaction would allow the authors to draw stronger conclusions and to add specifics to the summarizing model in Figure 6.

- With recombinant proteins in hand, the authors can test whether this physical interaction occurs in solution.
- To assess binding in the presence of MTs, can the authors look at HURP and Kif18 in a co-localization experiment on MTs. As in Figure 4A, but with lower concentrations?
- Do in silico predictions provide evidence for HURP-Kif18a interactions?

2)
Figure 5 contains crucial experiments with interpretations that return in the papers title and abstract. The co-localization experiments shown in panels 5G-I (and in Suppl. Figure 13) are especially nice and address the crux of the matter for this work. The overall message of the paper can therefore be improved by:
- showing more examples of data as in panels 5G and 5H (in the main and/or in the supplements)
- repeating these experiments with varying amounts of protein if possible
- quantifying the data shown in Suppl. Figure 13 and showing this as a main figure. These data do, after all, form the foundation for the final paragraph of the results section.

Moreover, the readability of Figure 5 (and Suppl. Fig. 13) should be improved at several points:

- please clearly indicate used concentrations and times of additions of fluorescent proteins and non-fluorescent proteins.

This information is important. But currently cryptically described in the legend and absent from the figure itself.

- please indicate where the MT seeds are positioned. E.g. in panel E, catastrophes do surprisingly not refer to the points in the kymograph where the MT switches from growing to shrinking. This is confusing.
- please show the interesting events (such as the release of Kif18a before the plus-end growing rate increases, the white arrow) with a magnification / improved spatial and temporal resolution

3)
The structure of HURP on microtubules is insightful and the structure determination from the co-decorated MTs is creative and impressive. Since HURP residues 87-132 could be identified, corresponding to MTBD1, this raises a question about the contribution of the other parts of HURP 1-285 to the interplay with Kif18a on MTs. How does MTBD2 contribute?

- To which degree does the MTBD1 encompassing fragment (105-150 as indicated, or 105-283, or e.g. the identified density 87-132) mimic the behavior of the HURP 1-285?

It is understandable if this cannot be addressed experimentally within the scope of this revision. If that is the case, please use the discussion to connect insights from the structure (and the HURP fragments not in there) with the rest of the experimental data.

- Please cite the recent work by the Zhang and Petry labs and compare structure and model with their data: HURP facilitates spindle assembly by stabilizing microtubules and working synergistically with TPX2
<https://www.biorxiv.org/content/10.1101/2023.12.18.571906v1>
BioRxiv, December 2023

Minor points

- For Figure 1A, please scale the cartoon to be approximately correct. Currently residues 401-846 are represented with the same length as residues 22-50
- The reader expects to see both HURP and Kif18a on MTs after seeing the reconstitution schematic with the different fluorophores in panel 1B. Is it possible to show the HURP signals for data shown in panel 1E and Figure 2?
- Please clearly indicate in Fig 1E and 1F that Kif18 is monitored and mention the concentration used.
- Please include a SDS-PAGE analysis of protein constructs used in this study in Figure 1 or as a supplementary figure.
- Please be consistent for the y-axes scaling throughout Figure 2 for a better comparison. Alternatively, keep the figure as it is but add a graph with curves from panels A-D. The arrangement of examples and data in Figure 2 might be improved by showing the examples of Kif18 mobility as a function of HURP titration horizontally instead of vertically.

Version 1:

Reviewer comments:

Reviewer #1

(Remarks to the Author)

I believe the authors have responded effectively to the questions, even conducting a new experiment to address the first concern. A new section, which includes a structural comparison of KIF18A with Kip3, is well-executed and offers valuable insights. I have two comments regarding the additional information:

1. The new Figure 3 includes AlphaFold3 models of complexes of KIF18A[1-480] and HURP[285-400]. The authors should add a small panel where the structure is color-coded by the confidence scores provided by the program, as is commonly done. This addition is necessary to assess the quality of the prediction in the areas of interest. Additionally, the single molecule data presented in Figure 3c is not quantified, unlike in the other figures. As it is, the level of evidence in this figure is lower than in the others ; it might be better suited for the supplementary material.

2. The authors added a new observation, reporting a 0.9% expansion of the microtubule lattice upon HURP binding to the microtubule (lines 171-174 and lines 530-534 in the Methods). The reference Taxol-stabilized microtubule map and model (EMDB/PDB) used as a reference and its resolution should be mentioned there ; if this is a new map/model they should be deposited.

Reviewer #2

(Remarks to the Author)

The authors have addressed all my concerns and I am happy to recommend this work for publication in Nature Communications.

Reviewer #3

(Remarks to the Author)

The authors have successfully addressed concerns raised and improved the manuscript.

The inclusion of the in silico predictions in the new main figure 3 requires a more rigour and information:

- include a prediction using AF2 multimer. AF2 multimer is the current standard for protein-protein predictions and more robust and reliable than AF3. This could replace the AF3 predictions or be appended (e.g. in a supplementary figure).
- include the pLDDT and pAE scores as quality predictions to appreciate these predictions.
- complete the methods section accordingly

I applaud the authors for their efforts during the revision and, after addressing the issue above, support the publication of this work.

We thank all reviewers for their positive assessment of our work and for the insightful comments they provided. In this point-by-point response, the reviewer comments are shown in italics and our responses are shown in blue.

REVIEWER COMMENTS

Reviewer #1 (Remarks to the Author):

Kinesin-8 are dual activity kinesins with both motile and microtubule depolymerization properties. They are important for the regulation of microtubule dynamics and spindle organization, and their activity needs to be regulated. During division in human cells, the kinetochore fibers recruit multiple proteins including the kinesin-8 Kif18A and Hepatoma upregulated protein (HURP) which is thought to modulate Kif18A activity. In this manuscript, the authors describe the complex regulation of human Kif18A by HURP. The demonstration includes a solid series of single molecule experiments performed on a range of constructs of both Kif18A and HURP. The single molecule work is complemented by a high quality cryo-EM analysis of the microtubule interaction with HURP alone or together with a monomeric Kif18A construct in AMPPNP. The single molecule experiments show a non-linear effect of HURP on Kif18A activity. The authors manage to rationalize mechanistically most of the single molecule results using the cryo-EM results, and provide a high-resolution structure of HURP interacting with a microtubule. The conclusions are solid, the manuscript is well written and well polished. Overall, this is an important contribution to the regulation of molecular motors and I'm very enthusiastic about this paper being published in Nature Communications once the comments below are addressed.

Note: In the document I received the line numbers are not unique: there is for instance a line 668 in the middle of both pages 34 and 35, so the comments below are given with both a page number and a line number.

Main comments:

1. On the representative images shown in Figure 1C for HURP, a substantial amount of fluorescence appears to originate from clusters (/aggregates/droplets). These clusters are possibly more prevalent in the images selected for the two longest constructs. Does Kif18A interact with these HURP clusters? Is it possible to rule out a scenario in which the longer the HURP construct the more HURP protein segregates into these clusters (to the expense of the HURP fraction that binds microtubule)? Such a scenario could for instance contribute to the trend seen in panel 1F – normalized run frequency (the longer the HURP construct the less Kif18A is activated/recruited).

We appreciate the reviewer's comment and recognize the concern raised. As illustrated in the following image, HURP aggregation did not significantly induce Kif18A colocalization. The majority of Kif18A remained associated with microtubules, despite the presence of substantial HURP aggregation in the background.

To address this concern, we improved the quality of the TIRF surface by adding a “piranha solution” cleaning step during coverslip preparation (lines 617-623) and optimized assay conditions to minimize clustering of HURP. We repeated the HURP binding and Kif18A motility experiments under these conditions and observed much fewer clusters in the background. We obtained similar results, indicating that the HURP clustering we observed previously in the background did not affect our conclusions (see new Figure 1 C).

2. page 6 - line 134-135 "No density corresponding to the HURP MTBD2 was present in the map". This statement should ideally be supported by showing in a supplementary figure a ~ 8 - 12 Å low passed filtered version of the reconstruction without any mask applied nor sharpening and displayed at low threshold (or something equivalent).

We generated Supplementary Figure 7 displaying the map at a low threshold and without dust hiding. Our maps are generated by the “local refinement” job in CryoSparc, which needs a mask as a mandatory input. To make sure we were not cutting off any putative density for MTBD2 we generated this map by using a dilated mask that extends significantly towards the lumen and past the outer surface of the microtubule. In this map, we didn’t observe any additional density that could correspond to MTBD2.

3. page 7-9 - lines 154-187. This part would benefit from a brief comment on the kinesin motor state as explained here. The reported Kif18A structure resolution (in the 3-4 Å range) is a considerable improvement over the previous microtubule bound Kif18A structures (reported to be around 5 Å ; Locke, J. et al, PNAS, 2017, PMID:29078367). The resolution obtained in the present work is similar to the one reported with a yeast kinesin-8 Kip3 (Hunter, B. et al, Nat Commun., 2022, PMID:35859148) obtained with a similar processing strategy. It would be informative to compare the conformation of the AMPPNP-bound Kif18A motor obtained by the authors with the ones of the latter paper, which reported in AMPPNP a motor conformation with a nucleotide binding pocket not closed and without the neck-linker docked. This way the reader would know with which conformation of this monomeric Kif18A motor (e.g. open/closed/semi-open) the cryo-EM experiment with Kif18A+HURP was done (since one can't strictly rule out that a different HURP behavior could occur with a different kinesin motor conformation). Secondly, this comment on the Kif18A motor state would also shed some light on Kif18A mechanochemistry and be of interest in the kinesin field.

We are very thankful to the reviewer for this constructive comment, as it made us look deeper into how our structure provides new insight on the mechanochemistry of Kinesin-8

motors. We added Supplementary Figure 13 with a structural comparison between Kip3 and Kif18A, highlighting differences in loop 2-microtubule interactions, closure of the nucleotide-binding pocket, and docking of the neck linker. The paragraph related to this figure can be found in lines 226-245 of the revised manuscript.

4. *The authors should explicit if the different single molecule traces used in a given plot originate from a single experiment (e.g. same protein prep and collected on the same microscope session) or from independent experiments, and in that case where the independence between experiments lies.*

We provided the number of technical replicates performed in the legends of Figures 1, 2, 6, and 7. Each technical replicate includes all experimental conditions represented in the corresponding plots, and consistent trends were observed across all replicates.

5. *Clarification of some physico-chemical conditions:*

- page 6 - lines 128-129 and page 32 - lines 644-645. *The expression "in the absence of added salt" is ambiguous and should be replaced by an expression that explicit the physico-chemical condition of the interaction (e.g. in buffer X at x°C).*

- *I have not seen any information regarding the temperature(s) used in the single molecule experiments. This/These temperatures should be stated (at least) in the material and methods ("Single-molecule motility imaging" and "Dynamic microtubule imaging", on page 25)*

We added the explicit physico-chemical conditions, including buffer composition and temperature, in the methods section (lines 636-641 and 647-650) and pointed to the methods section in the Results if the reader wants to know the exact buffer composition (see line 151).

6. *page 23 - line 519-520: "The GMPCPP microtubule seeds were usually good for 2 weeks." & page 24 - line 528-529 "The microtubule is stocked at room temperature and good to use for 2 weeks."*

Although these statements maybe useful to the readers, what is "good" is not defined and as importantly the statements do no clearly report what was done time wise with these seeds/microtubule. It would be preferable to state more what was done so that the reader can rule out any bias that could originate from variations in the materials (e.g. should different concentration conditions in a plot originate from fresh microtubules and others from 2 weeks old microtubules).

We corrected these statements in the methods section by using more objective language: "The GMPCPP microtubule seeds were kept at room temperature in the dark and used within 2 weeks." (lines 591-592) and "The taxol-stabilized microtubules were kept at room temperature in the dark and used within 2 weeks" (lines 601-602).

7. *page 28 - line 614: "Kd is determined from a fit to binding isotherm" and page 32 - line 646-647: "The fit used to determine the Kd is determined from a fit to binding isotherm"*

These informative fits - which are not mentioned in the material and methods - likely assume that each HURP molecule binds the microtubule independently (i.e. with no cooperativity).

Did the authors notice a significant difference in the helical parameters of the microtubule decorated with HURP compared to a naked microtubule of the same type stabilized similarly?

We now provide more detail on the fitting in the Methods section. We used a Hill equation where we fit both the K_d and the Hill coefficient n (lines 654-659). In addition to experimentally testing the binding of WT HURP¹⁻²⁸⁵ to microtubules, we added two new constructs with specific MTBD deletions (see Figure 1a). These experiments revealed a complex cooperativity behavior that likely involves structured and flexible elements both in tubulin and HURP. We also observe a slight compaction of the HURP-decorated microtubule lattice (compared to taxol-microtubules) (lines 171-174), which adds another layer of complexity to the observed cooperativity phenomena. These ideas are summarized in the new Supplementary Figure 10 and in the added text presented in the Results and Discussion sections (lines 180-188).

Minor comments:

page 3 - line 68: "to visualize how HURP and Kif18A bind to the microtubules at near-atomic resolution." "near-atomic resolution" being not well defined, it could be replaced by the actual resolution range of the complex.

We removed the “near-atomic resolution” expression.

page 16 - line 365: "minutes. microtubules were": capital letter needed for microtubules.

This was corrected (see line 429).

page 18 - line 401: "the length of an α - β tubulin dimer": the beta symbol doesn't have the same font as the other alpha/beta symbols.

This was corrected (see line 465).

page 24 - line 541: "Glass coverslips were coated s with polyethylene": I think there is a free "s" after coated.

This was corrected (see line 614).

page 25 - line 570: "containing 150 mM KAc and AMPPNP": the concentration of AMPPNP was omitted.

The concentration was added (2 mM) (see line 652).

page 28 - line 614: "S.D." and page 30 line 637: "S.E.". These common abbreviations should likely be explicited the first time they are used in the text.

This was corrected (line 696 and line 714).

page 33 - Fig 4 E. The relative orientation of the class averages could be indicated for non specialists (like what was done in panel D of that same figure).

The relative orientation of the classes was indicated (see revised Figure 5).

page 34 - line 668: "monoKif18A" is not defined (presumably Kif18A 1-373 / monomeric Kif18A).

This was corrected (see line 752).

page 42: The FSC curves plots in Supplementary Figure 5 are very small and hard to read. They should be scaled up.

This was corrected by scaling up the FSC images (see Supplementary Figure 6).

page 45 - line 753: "microtubule+HURP+Kif18A" dataset: use of "+" instead of "-" as in most of the paper, for instance in the line just above (752), this is written: "microtubule-HURP-Kif18A dataset". This occurs in several places and the use of "-" or "+" could be homogenized throughout the text.

The notation was unified by selecting the "-" character.

page 50: The headers of the supplementary table 1 should explicit which specific constructs of HURP and Kif18A were used.

This information was added to Supplementary Table 1.

Reviewer #2 (Remarks to the Author):

In the present manuscript, Perez-Bertoldi and co-workers investigate the molecular mechanism underlying the regulation of the kinesin-8 Kif18a by the Ran-GTP regulated MAP HURP. For some years now, these two proteins have a conspicuous enrichment at the chromosome proximal ends of kinetochore microtubules (kMTs) and have been implicated in chromosome congression by regulating kinetochore microtubule dynamics. However, the mechanism remained unclear, despite previous compelling in vivo evidence (mostly from Ye, F. et al. HURP Regulates Chromosome Congression by Modulating Kinesin Kif18A Function. Curr.Biol. 21 , 1584–1591 (2011)), showing that HURP overexpression inhibits Kif18a activity. By resorting to in vitro reconstitution analyses by TIRF microscopy and Cryo-EM studies, the authors now provide a near-atomic resolution explanation for how HURP regulates Kif18a activity. The key finding was that, at high concentration, HURP

competes with Kif18a for tubulin binding causing a steric clash. Interestingly, low HURP concentrations seem to activate Kif18a motility (through a HURP N-terminal activation domain), but this scenario seems unlikely in the context of their enrichment on kMTs and regulation of chromosome oscillations. The authors also report on possible Kif18a auto-inhibitory activity by its own C-terminal tail, similar to other kinesins. Another clarifying aspect is related with the controversial mechanism by which Kif18 regulates microtubule dynamics. Clearly, Kif18 is able to depolymerize stable microtubules (in the absence of free tubulin) on its own, but it does so rather inefficiently and in a manner that can be suppressed by HURP, supporting a microtubule capping model where Kif18a, assisted by HURP, suppresses microtubule plus-end dynamics to control kMT length. Overall, this is a technically very sound paper and the data are robust to sustain the proposed mechanism. I am therefore strongly supportive of publication of the present work and have only very minor textual suggestions, as detailed below:

1- Title: I find it misleading to include "mitotic spindle regulation" in the title of a paper that has no single experiment on mitotic spindles. I would recommend replacing by "microtubule length control".

We thank the reviewer for this suggestion. We accepted the reviewer's comment and replaced "mitotic spindle regulation" with "microtubule length control" in the title.

2- Introduction, page 2, line 52: it is yet unclear what the role of chromosome oscillations is and whether it serves any purpose in the alignment of chromosomes during mitosis. Without getting into detail on k-fiber length-dependent models of chromosome congression, a naïve reader might actually perceive that oscillations move chromosomes away from the equator, thus opposing chromosome alignment. I would either dwell a bit on models of chromosome congression or re-write the sentence for simplicity.

We removed the corresponding sentences in the first paragraph to avoid confusion, and only mention chromosome oscillations when we introduced Kif18A in the Introduction (see lines 58-60).

3- Results, page 6: Could the authors comment on the KD differences found for HURP binding to microtubules with or without physiological salt and discuss possible scenarios in which HURP might bind to microtubules without saturation (closer to physiological conditions) and respective implications for Kif18a activity/function?

In the results, we commented on the differences observed in affinity under different salt concentrations, likely due to the shielding effect of salt on protein-protein interactions (see lines 149-153).

In the discussion, we noted that HURP localizes in a concentration gradient across the K-fibers, which could provide physiological relevance to our concentration-dependent mechanism of Kif18A activation in the presence of HURP (see lines 308-313).

4- Discussion, page 12, line 272: yeast actually do not have a HURP orthologue. Thus, the “antenna model” might still hold true in yeast or even play a synergistic role with HURP in systems where HURP is present. This should be clarified.

This is a good point. We clarified this and added that, in HURP-containing organisms, both the “antenna model” and our proposed mechanism could be working synergistically to regulate Kif18A localization in K-fibers (see lines 327-333).

Reviewer #3 (Remarks to the Author):

Perez-Bertoldi and Zhao used state-of-the-art single-molecule and structural approaches to address the dynamics of Kif18a and HURP on microtubules. How microtubule-associated-proteins control the movement of motor proteins on microtubules is an interesting and important problem. Here, the authors focus on the interplay between HURP and Kif18a. Key advances from the paper are high-resolution structures of HURP and Kif18a on MTs and the concentration-dependent effects of HURP on Kif18a-microtubule interactions. At low concentrations, HURP promotes Kif18a motility. Probably due a release of Kif18a autoinhibition. At high HURP concentration the Kif18a motility is impaired. Probably due their MT binding in a competitive manner.

This is an interesting mode of action. Suppressed dynamics at the MT plus-end in the presence of HURP and Kif18a provides a clue for the regulation of spindle dynamics in vivo. Although implications of the proposed model on spindle length and k-fiber stability are not tested in cells, the in vitro experiments provide quantitative insights into HURP and Kif18a on microtubules under well-controlled conditions.

The authors addressed HURP-Kif18a / MAP-motor interplay on microtubules from a structural and single-molecule perspective, resulting in new insights that are of broad interest to the microtubule community. I therefore support the publication of this work and encourage the authors to address the following points:

Major points

1)

The authors speculate that the synergistic effects of HURP and Kif18a on stabilizing dynamics at the MT plus-end reflects an interaction between HURP (285-400) and Kif18a (1-480). This explains data in Figures 1 and 2 of the paper and is consistent with Figure 5 and Suppl. Figure 13. Direct evidence for this interaction would allow the authors to draw stronger conclusions and to add specifics to the summarizing model in Figure 6.

- With recombinant proteins in hand, the authors can test whether this physical interaction occurs in solution.

- To assess binding in the presence of MTs, can the authors look at HURP and Kif18 in a co-localization experiment on MTs. As in Figure 4A, but with lower concentrations?
- Do *in silico* predictions provide evidence for HURP-Kif18a interactions?

We thank the reviewer for this suggestion, as it helped us gather more evidence for the existence of this specific Kif18A-HURP interaction. AlphaFold 3 *in silico* predictions support an interaction between HURP³²¹⁻³³⁹ and the central β -blades of Kif18A. Additionally, we performed dual color imaging of HURP and Kif18A and observed Kif18A motors carrying HURP towards the microtubule plus-end. These results are consistent with direct interactions of HURP's "activating motif" and "decelerating motif" with Kif18A. These experiments are shown in the new main Figure 3 and in the main text referring to this figure (see lines 128-143).

2)

Figure 5 contains crucial experiments with interpretations that return in the papers title and abstract. The co-localization experiments shown in panels 5G-I (and in Suppl. Figure 13) are especially nice and address the crux of the matter for this work. The overall message of the paper can therefore be improved by:

- showing more examples of data as in panels 5G and 5H (in the main and/or in the supplements)

We show more examples of data in the new Figure 6, and in Supplementary Figure 16.

- repeating these experiments with varying amounts of protein if possible

We tested the effect of HURP and Kif18A on microtubule dynamics under higher concentrations of protein. We observed that the motor becomes so efficient at capping that the differences between the two HURP constructs are diluted. Therefore, we decided to continue with the reported concentrations as they clearly show a differential behavior between HURP¹⁻²⁸⁵ and HURP¹⁻⁴⁰⁰. In this study, we were not focused on how protein concentration affects microtubule dynamics, but rather show that HURP and Kif18A can have a joint effect in altering plus-end dynamics, which we think is a solid conclusion that follows from our data.

- quantifying the data shown in Suppl. Figure 13 and showing this as a main figure. These data do, after all, form the foundation for the final paragraph of the results section.

We included the quantification of the data in the new Figure 7e.

Moreover, the readability of Figure 5 (and Suppl. Fig. 13) should be improved at several points:

- please clearly indicate used concentrations and times of additions of fluorescent proteins and non-fluorescent proteins. This information is important. But currently cryptically described in the legend and absent from the figure itself.

We clarified the concentrations used for the dynamic microtubule experiments in the legends of Figures 6 and 7. All the proteins we used were fluorescently labeled and we made this explicit in the legends by adding the fluorescent label for each protein. We also added a sentence to the legend of these figures indicating how proteins were mixed and added to the flow chamber (see lines 785 and lines 802-803).

- please indicate where the MT seeds are positioned. E.g. in panel E, catastrophes do surprisingly not refer to the points in the kymograph where the MT switches from growing to shrinking. This is confusing.

We indicated the position of the seeds with a red line in Figures 6, 7 and in Supplementary Figure 16.

In the original manuscript, we defined catastrophes as shrinking events where the microtubule goes all the way back to the seed. In this revised manuscript, we changed the definition to any event where the microtubule transitions from growing to shrinking. Using this standard definition in the field, our data shows that HURP does not impact the catastrophe frequency, but increases the rescue frequency. This change was also incorporated into the manuscript (see lines 269-270 and lines 337-340).

- please show the interesting events (such as the release of Kif18a before the plus-end growing rate increases, the white arrow) with a magnification / improved spatial and temporal resolution.

We added magnified areas to mark the important events in Figures 6 and 7.

3)

The structure of HURP on microtubules is insightful and the structure determination from the co-decorated MTs is creative and impressive. Since HURP residues 87-132 could be identified, corresponding to MTBD1, this raises a question about the contribution of the other parts of HURP 1-285 to the interplay with Kif18a on MTs. How does MTBD2 contribute?

- To which degree does the MTBD1 encompassing fragment (105-150 as indicated, or 105-283, or e.g. the identified density 87-132) mimic the behavior of the HURP 1-285?

It is understandable if this cannot be addressed experimentally within the scope of this revision. If that is the case, please use the discussion to connect insights from the structure (and the HURP fragments not in there) with the rest of the experimental data.

To dissect the contribution of the different HURP elements to microtubule binding, we produced two new constructs with specific MTBD deletions and tested how their affinity for microtubules compares to that of WT HURP¹⁻²⁸⁵. Consistent with previous reports, the structured MTBD observed in the cryo-EM map is the main contributor to the affinity, while MTBD2 plays a secondary role that possibly involves flexible elements in tubulin (i.e. tubulin tails). This is summarized in Supplementary Figure 10 and explained in the corresponding text (see lines 180-188).

- Please cite the recent work by the Zhang and Petry labs and compare structure and model with their data:

HURP facilitates spindle assembly by stabilizing microtubules and working synergistically with TPX2

<https://www.biorxiv.org/content/10.1101/2023.12.18.571906v1>

BioRxiv, December 2023

We cited the work by Zhang and Petry, commented that our structural modeling is consistent with theirs, and mentioned their proposed model in our Discussion, in the context of K-fiber microtubule nucleation (see lines 346-348).

Minor points

- *For Figure 1A, please scale the cartoon to be approximately correct. Currently residues 401-846 are represented with the same length as residues 22-50*

We addressed this by making the segments to scale. Fixing a previous error, residue numbers corresponding to Kif18A's neck linker were changed from 363-373 to 353-370 in Figure 1a.

- *The reader expects to see both HURP and Kif18a on MTs after seeing the reconstitution schematic with the different fluorophores in panel 1B. Is it possible to show the HURP signals for data shown in panel 1E and Figure 2?*

We added an image showing the HURP signal for each construct to Figure 1e. We don't show the HURP signal in Figure 2 because we think it would make the figure unnecessarily crowded and now the HURP binding can be visualized in Figure 1, so we believe adding it also in Figure 2 would be redundant.

- *Please clearly indicate in Fig 1E and 1F that Kif18 is monitored and mention the concentration used.*

We added the Kif18A concentrations in Figure 1.

- *Please include a SDS-PAGE analysis of protein constructs used in this study in Figure 1 or as a supplementary figure.*

We added an SDS-PAGE gel as Supplementary Figure 1, with all the protein constructs used in this study.

- *Please be consistent for the y-axes scaling throughout Figure 2 for a better comparison. Alternatively, keep the figure as it is but add a graph with curves from panels A-D. The arrangement of examples and data in Figure 2 might be improved by showing the examples of Kif18 mobility as a function of HURP titration horizontally instead of vertically.*

We changed the Y-axis scales for all the plots to facilitate comparison between different conditions and displayed run frequency, velocity and run time plots horizontally instead of vertically.

We thank all reviewers for their final assessment of our work and for the insightful comments they provided. In this point-by-point response, the reviewer comments are shown in italics and our responses are shown in blue.

REVIEWER COMMENTS

Reviewer #1 (Remarks to the Author):

I believe the authors have responded effectively to the questions, even conducting a new experiment to address the first concern. A new section, which includes a structural comparison of KIF18A with Kip3, is well-executed and offers valuable insights. I have two comments regarding the additional information:

1. The new Figure 3 includes AlphaFold3 models of complexes of KIF18A[1-480] and HURP[285-400]. The authors should add a small panel where the structure is color-coded by the confidence scores provided by the program, as is commonly done. This addition is necessary to assess the quality of the prediction in the areas of interest.

We added Supplementary Figure 5 with the requested confidence metrics (prediction colored by pLDDT score and PAE matrix).

Additionally, the single molecule data presented in Figure 3c is not quantified, unlike in the other figures. As it is, the level of evidence in this figure is lower than in the others ; it might be better suited for the supplementary material.

We quantified the run frequencies for each condition and included the results in the new panel Figure 3c. Results are normalized to the run frequency experienced by HURP¹⁻⁸⁴⁶ in the presence of Kif18A.

2. The authors added a new observation, reporting a 0.9% expansion of the microtubule lattice upon HURP binding to the microtubule (lines 171-174 and lines 530-534 in the Methods). The reference Taxol-stabilized microtubule map and model (EMDB/PDB) used as a reference and it's resolution should be mentioned there ; if this is a new map/model they should be deposited.

We used a new map for a taxol-stabilized microtubule (3.3 Å) and its associated model, which we have deposited in the PDB/EMDB. We also added the EMDB and PDB accession codes in the data availability statement.

Reviewer #2 (Remarks to the Author):

The authors have addressed all my concerns and I am happy to recommend this work for publication in Nature Communications.

Reviewer #3 (Remarks to the Author):

The authors have successfully addressed concerns raised and improved the manuscript.

The inclusion of the in silico predictions in the new main figure 3 requires a more rigour and information:

- include a prediction using AF2 multimer. AF2 multimer is the current standard for protein-protein predictions and more robust and reliable than AF3. This could replace the AF3 predictions or be appended (e.g. in a supplementary figure).

We disagree with the statement that AF2 multimer is superior in performance to AF3, as the AF3 paper contains objective benchmarks showing better performance of AF3 in predicting protein-protein interactions (see reference 32 from our manuscript). Thus, we would like to move forward with the results we obtained using AF3.

- include the pLDDT and pAE scores as quality predictions to appreciate these predictions.

This has been incorporated in the new Supplementary Figure 5.

- complete the methods section accordingly

We now mention in the methods section that we include pLDDT and PAE scores.

I applaud the authors for their efforts during the revision and, after addressing the issue above, support the publication of this work.